# Bacterially produced metabolites protect *C. elegans* neurons from degeneration

**Arles Urrutia**[1,2°], **Víctor A. García-Angulo**[2,3°], **Andrés Fuentes**[2°],
**Mauricio Caneo**[1,2°], **Marcela Legüe**[1,2], **Sebastián Urquiza**[2], **Scarlett E. Delgado**[1,2],
**Juan Ugalde**[2], **Paula Burdisso**[4], **Andrea Calixto**[1,2]*

1 Centro Interdisciplinario de Neurociencia de Valparaíso, Facultad de Ciencias, Universidad de Valparaíso, Valparaíso, Chile, 2 Centro de Genómica y Bioinformática, Facultad de Ciencias, Universidad Mayor, Santiago de Chile, Chile, 3 Instituto de Ciencias Biomédicas, Facultad de Medicina, Universidad de Chile, Santiago de Chile, Chile, 4 Instituto de Biología Molecular y Celular de Rosario (IBR-CONICET), Facultad de Ciencias Bioquímicas y Farmacéuticas, Universidad Nacional de Rosario and Plataforma Argentina de Biología Estructural y Metabolómica (PLABEM), Rosario, Santa Fe, Argentina

☯ These authors contributed equally to this work.
* tigonas@gmail.com

**Data Availability Statement:** All relevant data are within the paper and its Supporting Information files.

## Abstract

*Caenorhabditis elegans* and its cognate bacterial diet comprise a reliable, widespread model to study diet and microbiota effects on host physiology. Nonetheless, how diet influences the rate at which neurons die remains largely unknown. A number of models have been used in *C. elegans* as surrogates for neurodegeneration. One of these is a *C. elegans* strain expressing a neurotoxic allele of the mechanosensory abnormality protein 4 (MEC-4d) degenerin/epithelial Na⁺ (DEG/ENaC) channel, which causes the progressive degeneration of the touch receptor neurons (TRNs). Using this model, our study evaluated the effect of various dietary bacteria on neurodegeneration dynamics. Although degeneration of TRNs was steady and completed at adulthood in the strain routinely used for *C. elegans* maintenance (*Escherichia coli* OP50), it was significantly reduced in environmental and other laboratory bacterial strains. Strikingly, neuroprotection reached more than 40% in the *E. coli* HT115 strain. HT115 protection was long lasting well into old age of animals and was not restricted to the TRNs. Small amounts of HT115 on OP50 bacteria as well as UV-killed HT115 were still sufficient to produce neuroprotection. Early growth of worms in HT115 protected neurons from degeneration during later growth in OP50. HT115 diet promoted the nuclear translocation of DAF-16 (ortholog of the FOXO family of transcription factors), a phenomenon previously reported to underlie neuroprotection caused by downregulation of the insulin receptor in this system. Moreover, a *daf-16* loss-of-function mutation abolishes HT115-driven neuroprotection. Comparative genomics, transcriptomics, and metabolomics approaches pinpointed the neurotransmitter γ-aminobutyric acid (GABA) and lactate as metabolites differentially produced between *E. coli* HT115 and OP50. HT115 mutant lacking glutamate decarboxylase enzyme genes (*gad*), which catalyze the conversion of GABA from glutamate, lost the ability to produce GABA and also to stop neurodegeneration. Moreover, in situ GABA supplementation or heterologous expression of glutamate decarboxylase in *E. coli* OP50 conferred neuroprotective activity to this strain. Specific *C. elegans* GABA transporters and receptors were required for full

**Funding:** Millennium Scientific Initiative of the Chilean Ministry of Economy, Development, and Tourism (P029-022-F) to AC, Proyecto Apoyo Redes Formacion de Centros (REDES180138) to AC, and CYTED grant P918PTE 3 to AC. MC received funding from Fondecyt 1181089 to AJM and AC. The funders had no role in study design, data collection and analysis, decision to publish, or preparation of the manuscript.

**Competing interests:** The authors have declared that no competing interests exist.

**Abbreviations:** ALM, anterior lateral microtubule; AVM, anterior ventral microtubule; AxT, truncated axon; AxW, wild-type axon; Axϕ, degenerated axon; CNS, central nervous system; DAF-16, ortholog of the Forkhead box transcription factor; DAF-2, ortholog of the insulin receptor; DEG/ENaC, degenerin/epithelial $Na^+$ channel; *deg-1*, degenerin-1; dsRNA, double-stranded RNA; F, filial generation; FOXO, Forkhead box; GABA, γ-aminobutyric acid; GABAse, GABA-aminotransferase plus succinic semialdehyde dehydrogenase; *gad*, glutamate decarboxylase enzyme gene; GAD, glutamate decarboxylase; GADT, glutamate/GABA antiporter; GFP, green fluorescent protein; IIS, insulin/IGF-1-like signaling; L4, fourth larval stage; LgFC, Log 2 Fold Change; MEC-4, mechanosensory ion channel subunit; MEC-4d, mechanosensory abnormality protein 4; NMR, nuclear magnetic resonance; OD, optical density; OPLS-DA, orthogonal projections to latent structures discriminant analysis; P, parental generation; PCA, principal component analysis; PD, Parkinson disease; PLM, posterior lateral microtubule; PVC, ventral cord interneuron; PVM, posterior ventral microtubule; rcs, regulator of capsular system; RNAi, RNA interference; ROS, reactive oxygen species; SSA, succinic semialdehyde; TM, transmembrane; TRN, touch receptor neuron.

HT115-mediated neuroprotection. Additionally, lactate supplementation also increased anterior ventral microtubule (AVM) neuron survival in OP50. Together, these results demonstrate that bacterially produced GABA and other metabolites exert an effect of neuroprotection in the host, highlighting the role of neuroactive compounds of the diet in nervous system homeostasis.

## Introduction

Intestinal microbes regulate many aspects of host physiology [1], including immune system maturation [2,3,4], neurodevelopment, and behavior [5,6,7], among others. Recent reports show that in mood disorders and neurodegenerative diseases, the microbiome composition and abundance is altered, and this has provided a glimpse at the role of specific bacterial metabolites with neuroactive potential in the prevention of such disorders [8,9,10,11]. However, whether bacterial metabolites directly influence neuronal degeneration and their mechanisms of action are largely unknown. The bacterivore nematode *Caenorhabditis elegans* continues to provide an excellent model to study the relationship between bacteria and host [12]. Both the animal and its bacterial diet are genetically tractable, making them suitable for individual gene and large-scale mutation analysis. This system has been instrumental in deciphering specific metabolites from gut bacteria that influence developmental rate, fertility and aging [13], and host factors mediating germline maintenance in response to a variety of bacterial diets [14] as well as defensive behavioral strategies against pathogens [15].

Genetically encoded prodegenerative stimuli, such as a dominant gain-of-function mutation on the mechanosensory channel gene *mec-4 (mec-4d)*, encoding the MEC-4d degenerin, have proven effective in deciphering common molecular players of neuronal degeneration in invertebrates and mammals [16,17,18]. The touch receptor neurons (TRNs) of *C. elegans* respond to mechanical stimuli by causing an inward $Na^+$ current through the MEC-4 channel, a member of the degenerin/epithelial $Na^+$ (DEG/ENaC) family. Mutations near the second transmembrane (TM) helix (A713), termed *mec-4d*, cause the constitutive opening of the channel and the degeneration of the TRN, rendering animals insensitive to touch [19]. Necrosis of the TRNs in *mec-4d* worms is presumably due to unregulated $Na^+$ and $Ca^{2+}$ entry as well as reactive oxygen species (ROS) imbalance [18,20,21]. The degeneration of the TRN is a stochastic process that begins with the fragmentation of the axon followed by the swelling of the soma. Notably, the use of this model has allowed for interventions that can delay neurodegeneration, such as caloric restriction, antioxidant treatment, and mitochondria blockage [16], as well as diapause entry [22]. In this study, we evaluated the rate of degeneration of *C. elegans* neurons in different dietary bacteria and found that specific dietary bacteria promote protection from neuronal degeneration. Combining systems biology approaches coupled to genetics, we discovered that γ-aminobutyric acid (GABA) produced by bacteria is protective for *C. elegans* neurons undergoing degeneration. Moreover, further characterization indicated that GABA was not the sole metabolite involved in neuroprotection, and lactate was also identified.

## Results

### Bacterial diet influences the rate at which neurons degenerate

We measured the effect of different dietary bacteria on the progression of genetically induced neuronal degeneration of the TRNs in a *C. elegans mec-4d* strain expressing a mutant

mechanosensory channel, MEC-4d [19]. We previously showed that *mec-4d*-expressing anterior ventral microtubule (AVM) touch neuron dies in a stereotyped fashion and defined the window of time when animals feed on the standard laboratory *E. coli* OP50 diet [16]. Right after hatching, *mec-4d* mutant animals were fed different bacteria, and the AVM neuronal integrity was quantified in adulthood (72 hours later). The pertinence of the use of each of these strains is explained in the Materials and methods section. The dietary bacteria used were *E. coli* OP50, *E. coli* B, *E. coli* HT115, *E. coli* K-12, *Comamonas aquatica*, *Comamonas testosteroni*, *Bacillus megaterium*, and the mildly pathogenic *Pseudomonas aeruginosa* PAO1. As a soil nematode, *C. elegans* feeds on a large range of bacteria in its natural environment [23]. We also selected three bacterial species previously coisolated with wild *C. elegans* from soil by our group, namely *Pseudochrobactrum kiredjianiae*, *Stenotrophomonas humi*, and *B. pumilus*. In accordance with our previous reports, neurodegeneration steadily occurred when feeding with *E. coli* OP50: only a very low percentage of worms (1%–3%) maintained AVM axons after 3 days (Fig 1A). Notably, although neurodegeneration occurred with *E. coli* B, *C. testosteroni*, *B. megaterium*, and *P. kiredjianiae* similarly to when feeding with *E. coli* OP50, the bacteria *E. coli* HT115, *E. coli* K-12, *C. aquatica*, *P. aeruginosa*, *S. humi*, and *B. pumilus* gave significant protection (Fig 1B). *E. coli* HT115 was the most protective, with over 40% of wild-type axons 72 hours after hatching, compared with less than 6% in *E. coli* OP50 (Fig 1B and 1D). The broad difference on neuronal integrity in *mec-4d* worm populations feeding on *E. coli* OP50 or HT115 can be observed in Fig 1C.

TRNs are neurons expressing receptors of gentle mechanical stimuli [24]. Hence, we determined the response to gentle touch in worms fed the different strains to test whether morphological protection shown in Fig 1B translates into functional responses. Fig 1E shows that the number of responses in worms correlates with the morphological categories AxW and AxL, the two axonal categories defined as functional in previous work [22].

## Bacterial components promote neuroprotection

Phenotypical outcomes mediated by intestinal bacteria can be a result of either a modulation of host physiology by interspecies live interactions (i.e. bacterial colonization) or by the exposure of the host to a bacterial metabolite. The first one requires bacteria to be alive in the intestine, whereas the second does not. To distinguish between these two possibilities, we fed *mec-4d* animals with UV-killed HT115 bacteria, the most protective among those tested, and scored the AVM integrity at 72 hours. Additionally, we also evaluated UV-killed *P. aeruginosa* PAO1, a mild pathogen that needs to be alive to colonize and induce long-term defensive responses in the animal [15]. In both cases, dead bacteria protected to the same extent as live bacteria (Fig 2A). This indicates that protective molecules of *E. coli* HT115 bacteria are produced prior to exposure to the animals, and thus neuronal protection is independent of the induction of bacterial responses by the interaction with the host. Furthermore, worms raised in dead HT115 bacteria cultivated to different optical densities (ODs) displayed the same levels of neuroprotection, suggesting that the protective factors are present in the bacteria during all phases of the growth curve (S1A and S1B Fig).

The large difference in neuroprotection between *E. coli* OP50 and HT115 strains raises two possibilities: (1) *E. coli* OP50 actively promotes the degeneration of the neuron, and (2) HT115 has a protective effect. To discern this matter, we fed worms with a mix of UV-killed *E. coli* HT115 and OP50 in different proportions (Fig 2B and 2C). A 1/100 (1%) dilution of *E. coli* HT115 in OP50 was sufficient to protect AVM neurons significantly more than pure OP50. This strongly suggested that *E. coli* HT115 produces a neuroprotective compound that is needed in small amounts.

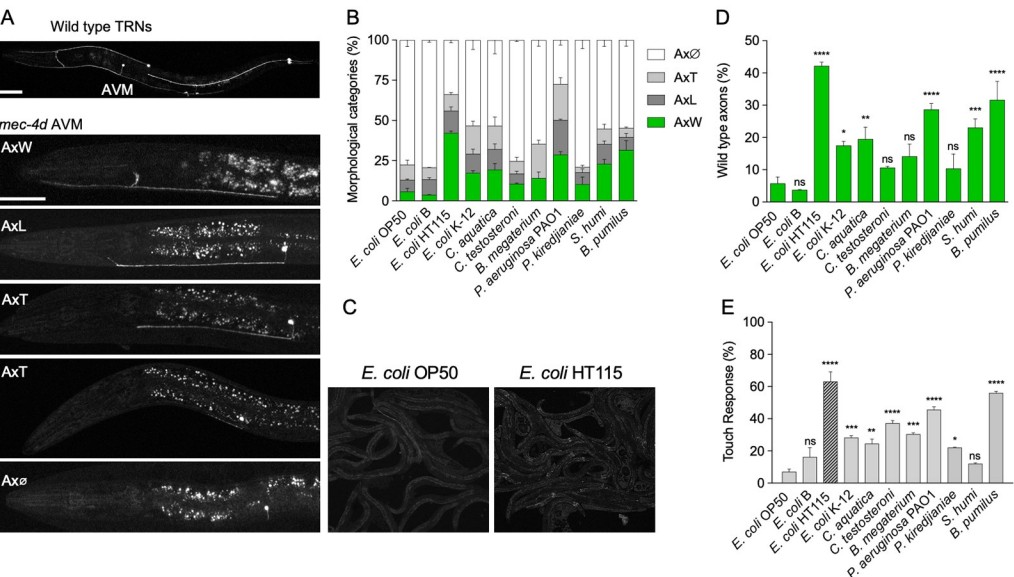

**Fig 1. Dietary bacteria determine the rate of neuronal degeneration.** (A) TRNs expressing GFP in wild-type and *mec-4d* worms. The latter shows the stereotypical progression of AVM degeneration of *mec-4d* mutants and constitutes the axonal categories assessed during the experiment shown in (B). Scale bars represent 20 μm. (B) Percentage of all morphological axonal categories in *mec-4d* worms after 72 hours of growth in different bacterial strains. (C) Fluorescence microscopy fields of GFP-expressing *mec-4d* worms raised in the indicated *E. coli* strains comparing the presence of AVM axons on each preparation at 10× magnification. (D) Percentage of wild-type axons in the experiment shown in (B). (E) Percentage of touch responsiveness of animals after growth in the different dietary bacteria. $^{****}P < 0.0001$, $^{***}P < 0.001$, $^{**}P < 0.005$, $^*P < 0.05$, ns. The underlying numerical data and statistical analysis for each figure panel can be found in S1 and S2 Datasets, respectively. AVM, anterior ventral microtubule; Axϕ, degenerated axon; AxT, truncated axon; AxW, wild-type axon; GFP, green fluorescent protein; *mec-4d*, mechanosensory abnormality protein 4; ns, not significant; TRN, touch receptor neuron.

To test whether the neuroprotective molecules are being secreted by the bacteria, we separated the supernatant of both bacterial strains from their pellets by centrifugation and mixed the supernatant of *E. coli* HT115 with OP50 pellet and vice versa (see Materials and methods for details). *E. coli* HT115 supernatant was not capable of providing protective activity when

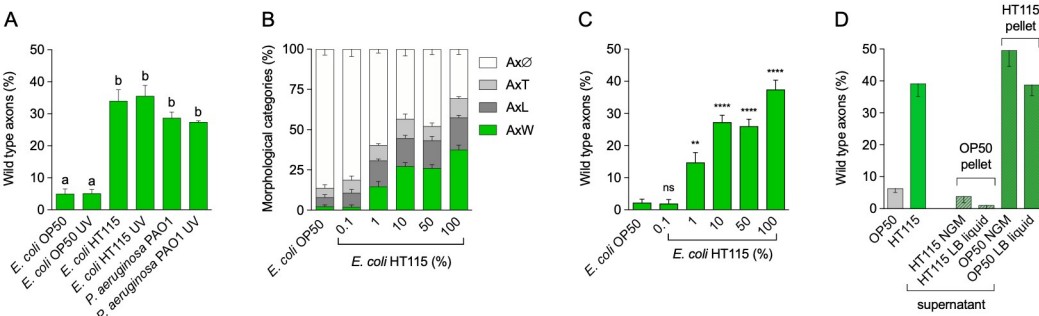

**Fig 2. Bacterial components have neuroprotective activity.** (A) Axonal categories in worms raised in the indicated live or UV-killed bacterial strains. (B–C) All axonal categories (B) or wild-type (C) axons in worms rose in different proportions of UV-killed *E. coli* HT115 on UV-killed OP50. (D) Wild-type axons of *mec-4d* animals fed with *E. coli* HT115 or OP50 bacterial pellet supplemented with supernatant of OP50 or HT115. $^{****}P < 0.0001$, $^{***}P < 0.001$, $^{**}P < 0.005$, $^*P < 0.05$; ns. "a" and "b" are used to indicate statistically significant differences. The underlying numerical data and statistical analysis for each figure panel can be found in S1 and S2 Datasets, respectively. Axϕ, degenerated axon; AxT, truncated axon; AxW, wild-type axon; LB, Luria-Bertani; *mec-4d*, mechanosensory abnormality protein 4; NGM, nematode growth media; ns, not significant.

mixed with *E. coli* OP50 pellet (Fig 2D). This suggests either that the protective factor is not secreted or that the amounts contained in the supernatant are not sufficient for protection. As expected, *E. coli* OP50 supernatant did not alter the protection pattern of *E. coli* HT115.

## *E. coli* HT115 diet promotes long-term protection of mechanoreceptors and interneurons of the touch receptor circuit

*E. coli* HT115 was shown to be neuroprotective throughout the development of the animal and into young adulthood (Fig 1B). We explored whether AVM neurons are still protected after worms reached maturity. To that end, we fed newly hatched *mec-4d* animals with *E. coli* HT115 and scored their neuronal integrity every 24 hours for 168 hours. While on *E. coli* OP50, all animals had degenerated neurons at the final time point, and on HT115 food, 25% of animals had wild-type AVM axons (Fig 3A and 3C), confirming the ability of HT115 to significantly protect at later life stages. Notably, between 12 and 24 hours after hatching in HT115, there was a statistically significant rise in wild-type axons (AxW, Fig 3C), suggesting that neurons could be growing after an initial truncation. To assess this, we followed individual animals in a longitudinal fashion on *E. coli* HT115 and scored the neuronal integrity of each nematode every 24 hours for 3 days. We scored axons separately according to their initial and final morphology and classified axonal outcome as "protection" when the morphology of axons did not change from truncated or wild type and "degeneration" when axonal morphology changed from truncated axon (AxT) to degenerated axon (Axϕ) or was maintained as Axϕ. Finally, "regeneration" refers to axon growth from truncations to wild type. Although the most prevalent category is protection (40%), 30% of axons regenerated between 24 and 72 hours after

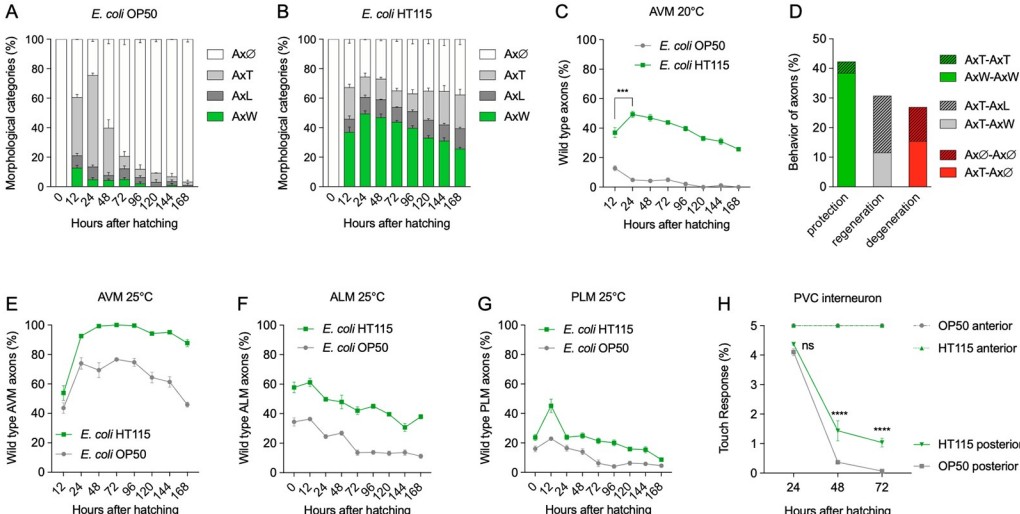

**Fig 3. Neuroprotection induced by dietary *E. coli* HT115 is long lasting and extends to other neuronal types.** Time course of axonal categories of worms fed with *E. coli* OP50 (A) or HT115 (B) for 168 hours. (C) Percentage of wild-type axons in (A) and (B). (D) Proportion of axonal categories in longitudinal assays. Protection (green) indicates axons that were not degenerated over time regardless of the initial category, with the exception of Axϕ. Regeneration (gray) accounts for axons that grew in size over time. Degeneration (red) accounts for axons that degenerated over time. Determination of wild-type axons in AVM (E), ALM (F), PLM (G), and PVC (H) neurons of worms fed *E. coli* OP50 or HT115. ****$P < 0.0001$, ***$P < 0.001$, ns. The underlying numerical data and statistical analysis for each figure panel can be found in S1 and S2 Datasets, respectively. ALM, anterior lateral microtubule; AVM, anterior ventral microtubule; Axϕ, degenerated axon; AxT, truncated axon; AxW, wild-type axon; ns, not significant; PLM, posterior lateral microtubule; PVC, ventral cord interneuron; PVM, posterior ventral microtubule.

hatching on *E. coli* HT115 (Fig 3D). This suggests that under HT115 protective conditions, a portion of neurons can repair broken axons.

Next, we explored whether other neurons of the touch circuit are protected from degeneration in *E. coli* HT115 diet. It has been already reported that at hatching, four embryonic TRN, two anterior lateral microtubule (ALMs), and two posterior lateral microtubule (PLM) neurons have already degenerated when growing at 20 ˚C in this model [16,25]. At 25 ˚C, however, degeneration proceeds at a slower rate [16,22]. To analyze the degeneration rate of ALM and PLM neurons, animals in the fourth larval stage (L4) were grown at 25 ˚C, and their progenies were synchronized at birth. The neuronal integrity of ALM, PLM, and AVM cells was assessed at 12, 24, and every 24 hours after birth until 168 hours at 25 ˚C. The percentage of wild-type neurons in *E. coli* HT115 diet is significantly higher than that in the OP50 diet throughout the temporal course for all three neurons (Fig 3E and 3G, full morphological characterization is shown in S2A and S2F Fig).

We tested next whether *E. coli* HT115 was capable of protecting the ventral cord interneuron (PVC) expressing the degenerin-1 (*deg-1*) prodegenerative stimulus. *deg-1(u38)* animals progressively lose the ability to respond to posterior touch due to the time-dependent degeneration of the PVC interneuron [26]. We tested the posterior touch response of *deg-1* animals during development feeding on *E. coli* OP50 and HT115. *E. coli* HT115 promotes a larger functional response than *E. coli* OP50, suggesting that this neuron is also protected (Fig 3H). Taken together, these results demonstrate that HT115 diet is protective over different neuronal types undergoing degeneration.

Neurodegeneration of the TRNs is directly related to the expression of a neurotoxic form of the mechanosensory channel. Therefore, it is formally possible that a decrease in the expression of the channel would diminish the prodegenerative stimulus and promote protection. We sought to evaluate whether HT115 diet changes the expression of the MEC-4d channel in the membrane. We constructed a double mutant of *mec-4d* expressing MEC-4::green fluorescent protein (GFP) and quantified the number of channels of PLMs in HT115-fed animals compared with OP50. S3A and S3B Fig show that channel number remains constant in both diets, ruling out that protection conferred by HT115 affects expression of MEC-4d channel in the membrane.

## Early exposure of animals to *E. coli* HT115 is sufficient for neuronal protection

Ad libitum feeding on *E. coli* HT115 protected *mec-4d*-expressing neurons from degeneration for long periods of time. We sought to investigate whether a constant stimulus provided by the HT115 metabolite is required to achieve neuroprotection or an early, discrete time-lapse exposure to the diet is sufficient. We fed animals for the first 6 hours after hatching (previous to the birth of the AVM) and for 12 hours after hatching (at birth of the neuron) with UV-killed *E. coli* HT115 and immediately switched to *E. coli* OP50. We scored the neuronal morphology 12, 24, 48, and 72 hours posthatching (S4A and S4D Fig). In parallel, both diets were fed ad libitum as controls. Strikingly, animals that ingested *E. coli* HT115 for only 6 hours showed a significantly larger number of wild-type neurons at 72 hours (14.3%) than animals continuously fed OP50 (3.6%, Fig 4A) and had more axons in the other categories (S4A and S4C Fig). Feeding HT115 to *mec-4d* animals for the first 12 hours after hatching conferred a significantly larger protection than feeding HT115 for only 6 hours (Fig 4A). These results show that although early short exposures are not equally protective as a permanent HT115 diet, they do have a long-lasting effect in neurons compared with an uninterrupted diet of *E. coli* OP50.

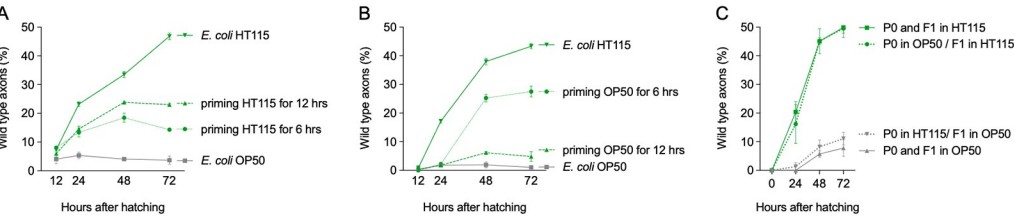

**Fig 4. Diet of *E. coli* HT115 at early stages of life is necessary and sufficient to confer neuroprotection.** (A and B) Percentage of wild-type axons of animals fed for 6 and 12 hours with (A) *E. coli* HT115 or (B) *E. coli* OP50 and then changed to OP50 or HT115 diet, respectively. (C) Percentage of wild-type axons of animals feeding on either *E. coli* OP50 or HT115 whose parents were fed on either diet. The underlying numerical data and statistical analysis for each figure panel can be found in S1 and S2 Datasets, respectively. F, filial generation; P, parental generation.

We then tested the effect of early exposure to nonprotective bacteria. We fed *mec-4d* animals for 6 and 12 hours with UV-killed *E. coli* OP50 and then changed them to HT115. The morphology of AVM neurons was scored at 12, 24, 48, and 72 hours posthatching. Six hours of *E. coli* OP50 exposure did not prevent HT115 from protecting AVM neurons later in adulthood (Fig 4B and S4E Fig). Exposure for 12 hours, however, precludes protection of the AVM (Fig 4B and S4F Fig). This suggests that the time between the first 6 and 12 hours of development is crucial for the protective effect to take place.

In *C. elegans*, some dietary bacteria–induced traits show heritable properties [15]. Therefore, we tested whether neuronal protection could be inherited. Animals were fed either *E. coli* OP50 or HT115 from birth, and their F1 progeny transferred to OP50 or HT115. Neuronal integrity of descendants was tested in a time course fashion. One generation of parental feeding on HT115 did not improve neuronal protection in the progeny feeding on OP50, nor did *E. coli* food preclude protection of filial generation (F) 1 feeding on HT115 (Fig 4C). This result indicates that the protective effect of *E. coli* HT115 is not transmitted intergenerationally.

## Identification of uniquely expressed genes on neuroprotective bacteria

To identify the bacterial molecule(s) conferring neuroprotection, we looked for differences in the genomes and transcriptomes of the two *E. coli* strains. We reasoned that genes important for neuronal protection would be uniquely expressed or up-regulated in *E. coli* HT115 compared with *E. coli* OP50. We first sequenced the genomes of *E. coli* HT115 and OP50 using the Illumina MiSeq platform. Notably, we found that *E. coli* OP50 has a deletion of 23 Kbp, containing genes for the regulators of capsular system (*rcsDB*, *rcsC*) required for envelope stress response, genes for short-chain polyhydroxybutyrate synthesis (*atoSC*, *atoDAEB*), and the pseudogene *yfaATSQP* [27]. Thus, *RcsB* is a unique *E. coli* HT115 gene that codes a main transcription factor required for the activation of the glutamate decarboxylase enzyme gene (*gad*) operons, alone or coupled to other regulators by positive feedback as illustrated in Fig 5A.

Transcriptomic analysis determined that the genes involved in resistance to acidic environments were highly up-regulated in *E. coli* HT115 (Fig 5B). The glutamate decarboxylase operons *gadAX* and *gadBC* include the genes *gadA* and *gadB*, both encoding for the enzyme glutamate decarboxylase (GAD), which converts glutamate to GABA, whereas *gadC* encodes a glutamate/GABA antiporter. Other overexpressed genes in HT115 include those for the periplasmic acid stress chaperones *hdeA* and *hdeB*. Importantly, this overexpression occurs while global regulators also involved in the acidic response remain equally expressed in both strains (CRP-AMPc, H-NS, or Fis). Therefore, *rcsB* deletion seems to induce a specific metabolic difference between both bacteria, i.e., the abolition of GABA production in *E. coli* OP50. To endure this deficit, genes related to sodium/glutamate and glutamate/aspartate transport were

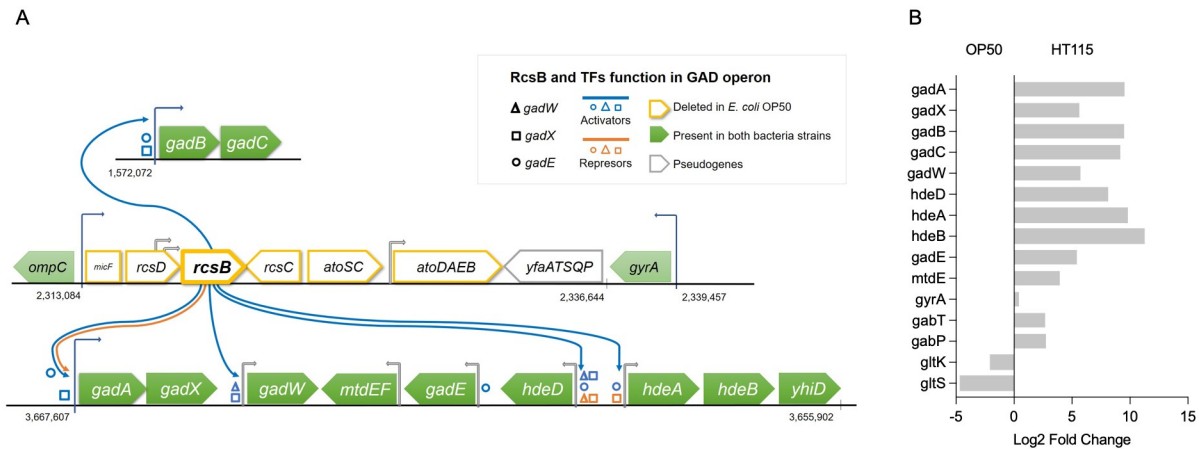

**Fig 5. Enzymes of the GAD enzyme operon are uniquely and highly expressed in *E. coli* HT115.** (A) Scheme of regulation of the *gad* operon. (B) Log2 Fold Change of genes coding for enzymes of GABA metabolism. The underlying numerical data and statistical analysis for each figure panel can be found in S1 Dataset and S2 File, respectively. GABA, γ-aminobutyric acid; GAD, glutamate decarboxylase; *gad*, glutamate decarboxylase enzyme gene; *rcs*, regulator of capsular system.

up-regulated in *E. coli* OP50 as shown in Fig 5B (*gltS* Log 2 Fold Change (LgFC) = −4.69, *gltK* LgFC = −2.08, respectively). Additionally, other genes related to GABA metabolism (the trans-aminase *gabT* LgFC = 2.67) and membrane permeability (permease *gabP* LgFC = 2.74) were also up-regulated in *E. coli* HT115 from non-*rcsB*-dependent operons. Interestingly, no other metabolic enzyme-related genes were up-regulated in either bacterium. This suggests that enzymes and metabolites involved in the pathway of GABA production and utilization are good candidate neuroprotective players.

## GAD and its product GABA are required for *E. coli* HT115 neuroprotection

To test the role of GAD and its product GABA in neuroprotection, we first generated a *gad* null mutant of *E. coli* HT115 by homologous recombination (HT115Δ*gad*, details in Materials and methods). To corroborate that HT115Δ*gad* lacked GAD activity, we used a colorimetric assay based on pH elevation given by the conversion of glutamate to GABA [28,29]. As expected, wild-type *E. coli* HT115 raised the pH of the solution, whereas neither HT115Δ*gad* nor OP50 were able to do so. To confirm that a rise in pH is due to the expression of GAD, we transformed *E. coli* OP50 with a plasmid expressing *gadA* (pG*gadA*). *E. coli* OP50 pG*gadA* supplemented with glutamate showed potent enzymatic activity, raising the pH of the solution above HT115 levels (Fig 6A). Secondly, we fed *mec-4d* animals with *E. coli* HT115Δ*gad* and scored its protective potential at 72 hours. HT115Δ*gad* was not able to protect degenerating AVM neurons, showing a significant reduction of wild-type axons compared with wild-type strains (Fig 6B). This shows that GAD activity plays a pivotal role in the protection conferred by HT115 bacteria. Moreover, plasmid pG*gadA* was able to rescue protective potential in null mutant HT115Δ*gad*. Additionally, dietary supplementation of UV-killed HT115Δ*gad* with 2 mM of GABA was sufficient to provide neuroprotection (Fig 6B and S5A Fig).

Finally, we fed *mec-4d* with *E. coli* OP50 pG*gadA* to test whether *gadA* is sufficient to provide protective activity in the presence or absence of glutamic acid, the substrate for GAD. Whereas *E. coli* OP50 pG*gadA* alone was not sufficient to increase wild-type axons incidence, glutamate addition to the bacterial culture significantly increased the presence of wild-type axons compared with *E. coli* OP50 pG*gadA* and *E. coli* OP50 wild type (Fig 6C and

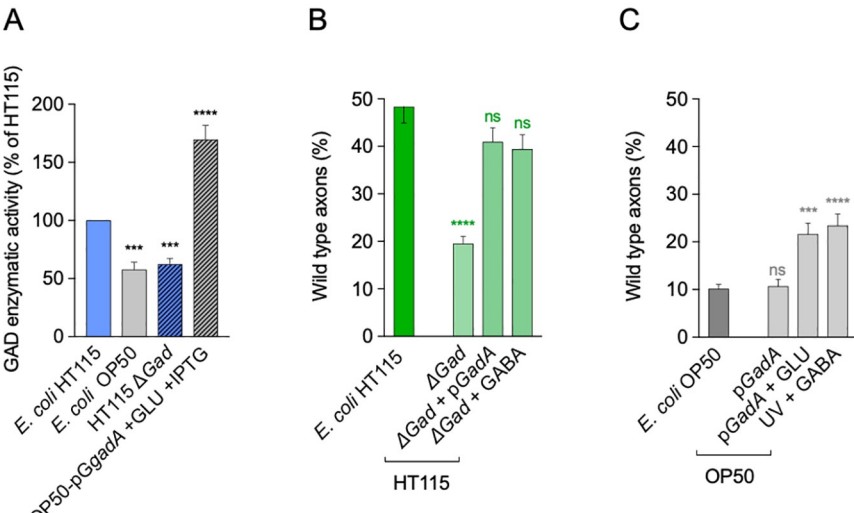

**Fig 6. Bacterial GABA is crucial for neuroprotection.** (A) Measurements of GAD enzyme activity normalized as a percentage of HT115 GAD activity in wild-type, mutant, and transformed bacterial strains. (B and C) Percentage of wild-type axons in wild-type and Δ*gad* mutant HT115 strain (B) and wild-type OP50 strain (C) supplemented with GABA and genetically transformed with the pG*gadA* plasmid. ****$P < 0.0001$, ***$P < 0.001$, ns. The underlying numerical data and statistical analysis for each figure panel can be found in S1 and S2 Datasets, respectively. GABA, γ-aminobutyric acid; GAD, glutamate decarboxylase; *gad*, glutamate decarboxylase enzyme gene; ns, not significant.

S5B Fig). This is coherent with the increased GAD activity of *E. coli* OP50 supplemented with pG*gadA* shown in Fig 6A. Furthermore, supplementation of HT115Δ*gad* with 2 mM GABA was sufficient to provide neuroprotection (Fig 6C). Importantly, addition of 2 mM GABA to UV-killed *E. coli* OP50 lawn protected *mec-4d* neurons significantly more than OP50 alone, even though it did not reach HT115 levels (Fig 6C). Taken together, these results show that GAD and its product GABA play an important role in *E. coli* HT115–mediated neuroprotection.

## Identification of metabolites expressed in neuroprotective conditions

To unbiasedly identify potentially neuroprotective metabolites produced by the strain HT115 but absent in the nonprotective HT115 Δ*gad* and *E. coli* OP50 strains, we implemented a non-targeted metabonomics approach using 1H nuclear magnetic resonance (NMR). A total of 24 extract samples were prepared and analyzed (eight of each strain). To evaluate the global metabolic profile of the three bacterial strains, we performed a principal component analysis (PCA) of binned 1H NMR datasets. As we expected, all three bacteria strains were metabolically different, with *E. coli* HT115 and HT115 Δ*gad* closer than *E. coli* OP50 in the metabolic space (S5A Fig). Metabolites related with neuroprotection were evaluated by orthogonal projections to latent structures discriminant analysis (OPLS-DA), first comparing *E. coli* HT115 strain against HT115 Δ*gad* (Fig 7A), and secondly, comparing *E. coli* HT115 against HT115 Δ*gad* and OP50 (Fig 7D). OPLS-DA models were validated by 200 permutations (S6B and S6C Fig). Discriminant analysis in HT115 versus HT115 Δ*gad* revealed intergroup metabolic differences. The discriminant up-regulated metabolites in wild-type HT115 were GABA, lactate, sucrose, and maltose, whereas in HT115 Δ*gad*, they were glutamate and putrescine (Fig 7B and 7C, S1 Table and S7 Fig). This is coherent with the accumulation of the GAD substrate glutamate, given the absence of the enzyme on HT115 Δ*gad*. Notably, the comparison between *E. coli*

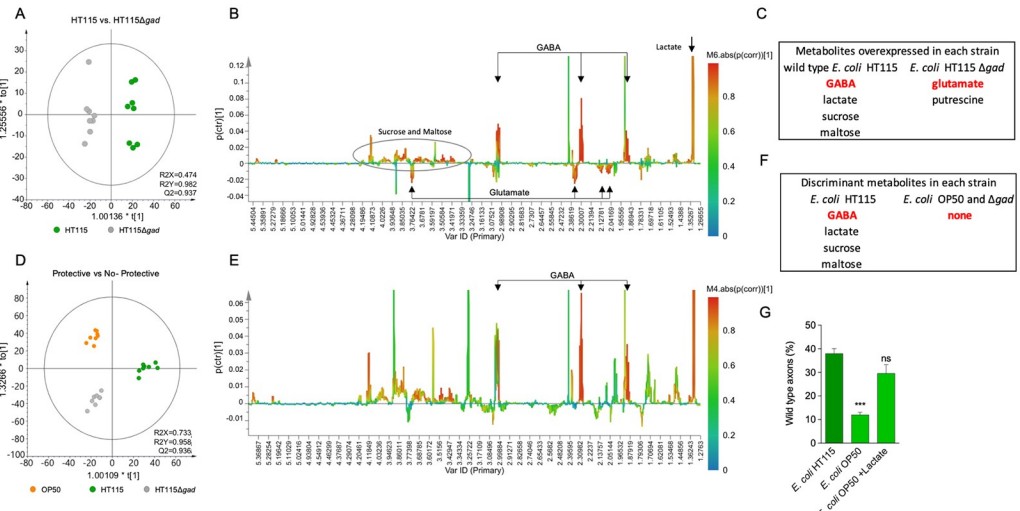

**Fig 7. Metabolomics analysis of neuroprotective and nonprotective bacteria.** (A–D) OPLS-DA score plot of protective
*E. coli* HT115 wild type (blue) and nonprotective *E. coli* HT115Δ*gad* (light gray) (A) and *E. coli* HT115 wild type (blue)
and nonprotective *E. coli* HT115Δ*gad* (light gray) and OP50 (dark gray) (D). (B–E) OPLS-DA S-line plots with pairwise
comparison of data from NMR spectra obtained comparing the *E. coli* HT115 strain against HT115 Δ*gad* (B) and the *E.
coli* HT115 strain against HT115 Δ*gad* and *E. coli* OP50 (E). Colors are associated with the correlation of metabolites
characterizing the 1H NMR data for the class of interest, with the scale shown on the right-hand side of the plot. In (B),
GABA and glutamate signals are shown. (C–F) Tables indicate which metabolites are differentially expressed in each
strain. (G) Percentage of wild-type axons in *mec-4d* animals fed with *E. coli* OP50 supplemented with lactate.
***P* < 0.001; ns. The underlying numerical data for the figure panels can be found in S1 and S2 Tables (A to F) and S1
Dataset (G). Statistical analysis can be found in S2 Dataset. GABA, γ-aminobutyric acid; *gad*, glutamate decarboxylase
enzyme gene; *mec-4d*, mechanosensory abnormality protein 4; NMR, nuclear magnetic resonance; ns, not significant;
OPLS-DA, orthogonal projections to latent structures discriminant analysis.

HT115 strain against HT115 Δ*gad* and *E. coli* OP50 revealed the following intergroup meta-
bolic differences: GABA, lactate, sucrose and maltose were highly expressed in the neuropro-
tective strain, whereas there were no discriminant metabolites found in higher levels in OP50
and HT115 Δ*gad* (Fig 7E and 7F and S2 Table). Overall, in strong agreement with our genetic
and chemical complementation approach, these results further indicated that GABA is one of
the metabolites playing a central role in HT115-conferred neuroprotection.

Importantly, the supplementation of *E. coli* OP50 with GABA did not reach HT115 neuro-
protective levels. We wonder whether other metabolites overabundant in *E. coli* HT115 could
also contribute to neuronal protection. Lactate is a metabolite not expressed either in the Δ*gad*
or OP50 bacteria. To test the role of lactate in neuroprotection of *C. elegans* TRNs, we fed ani-
mals with *E. coli* OP50 bacteria supplemented with 2 mM lactate. Fig 7G and S8 Fig show that
lactate confers significant protection to *E. coli* OP50 bacteria. This suggests that lactate in addi-
tion to GABA contribute to neuroprotection.

## GAD activity and GABA levels correlate with neuroprotection conferred by
other bacteria

The amount of a neuroprotective metabolite present in a given bacterium could be an indica-
tor of that bacteria's ability to confer neuronal protection. Bacteria tested by us in Fig 1B gave
different degrees of protection to *mec-4d* animals. We evaluated GAD activity in all strains and
normalized it against *E. coli* HT115 (Fig 8A). All strains except *E. coli* K-12 exhibited less GAD
activity than *E. coli* HT115. Additionally, we directly measured GABA production using the

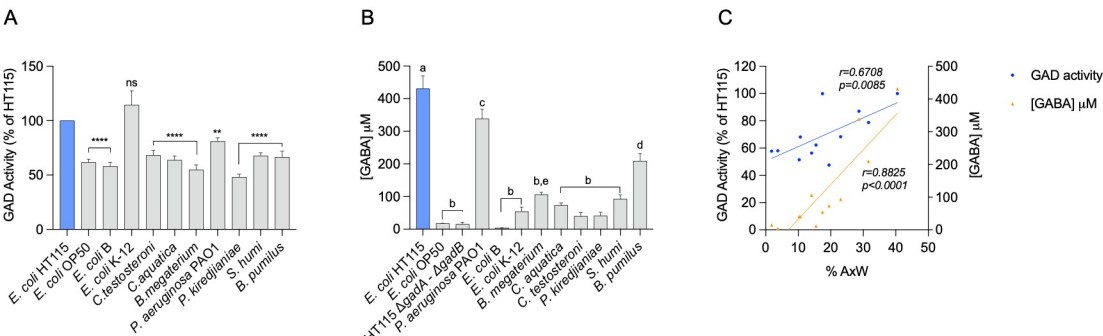

**Fig 8. GAD activity and GABA levels correlate with neuroprotection conferred by other bacteria.** Measurements of GAD enzyme activity normalized as a percentage of HT115 GAD activity (A) and GABA concentration found in pellets (B) in all bacteria used. (C) Correlation between GAD activity and GABA concentration in bacteria diet and percentage of wild-type axons in *C. elegans*. ****$P < 0.0001$, **$P < 0.005$; ns. "a," "b," "c," and "d" are used to indicate statistically significant differences. The underlying numerical data and statistical analysis for each figure panel can be found in S1 and S2 Datasets, respectively. GABA, γ-aminobutyric acid; GAD, glutamate decarboxylase; ns, not significant.

GABA-aminotransferase plus succinic semialdehyde dehydrogenase (GABase test [30] and S9 Fig). *E. coli* HT115 pellet had the highest GABA levels, whereas OP50 and HT115Δ*gad* were indistinguishable from each other (Fig 8B). This demonstrates that GABA is being produced in *E. coli* HT115 and not in OP50 or the HT115 Δ*gad* strain. *P. aeruginosa* PAO1 and *B. pumilus* had less GABA than HT115 but significantly more than most strains (Fig 8B). To understand whether there was a correlation between GAD and GABA levels with neuroprotection conferred by these bacteria, we performed a Pearson correlation test. Fig 8C shows that GAD expression and GABA levels are correlated with neuroprotective activity in all strains. Importantly, GABA concentration was a better indicator ($r = 0.88$) than GAD activity ($r = 0.67$). These results support the previous evidence that bacterial GAD enzyme and its product GABA are key for neuroprotection.

## Host GABA receptors and transporters are required for full HT115 bacteria–mediated neuroprotection

To discern whether systemic or neuronal GABA transport was implicated in neuroprotection, we silenced the expression of a number of candidate solute transporters (*unc-47*, *snf-5*) and GABA receptors (*gab-1*, *lgc-37*, *unc-49*) using RNA interference (RNAi). To distinguish a systemic from a touch cell–specific requirement for these effectors, we used *mec-4d* animals (in which neuronal RNAi is inefficient) and *mec-4d* animals sensitive to RNAi only in the TRN (WCH6, [16]). We fed double-stranded RNA (dsRNA) of the selected effectors to both strains, and the neuronal morphology was assessed at 72 hours. *lgc-37*, *snf-5*, *unc-47*, and *gab-1* dsRNA-expressing bacteria caused a discrete but significant decrease in neuronal protection in the systemic RNAi strain but not in the TRN-specific strain (Fig 9A and 9B and S10A and S10B Fig), suggesting these genes act in nonneuronal tissues to mediate neuroprotection.

## *E. coli* HT115 protection requires DAF-16 signaling

We explored the role of the insulin/IGF-1-like signaling (IIS) pathway, a well-described and conserved signaling cascade acting systemically. In *C. elegans*, down-regulation of the insulin receptor DAF-2 is neuroprotective [16]. We investigated whether the *E. coli* HT115 neuroprotective effect also involved the IIS. First, we fed *daf-2ts; mec-4d* animals with *E. coli* HT115 at 25 °C. This strain expresses a DAF-2 protein version that is unstable at 25 °C. Next, we scored

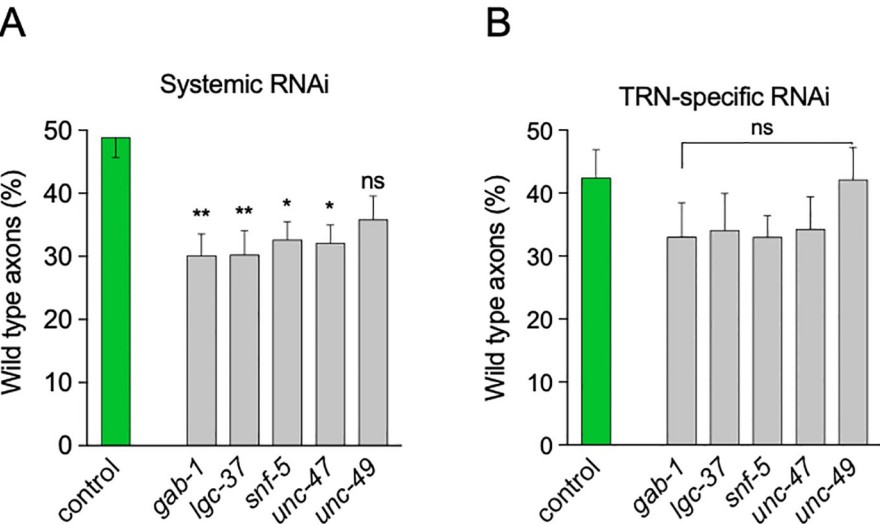

**Fig 9. Effect of silencing GABA effectors in *C. elegans* on neuroprotection.** Morphological integrity of AVM in animals feeding on dsRNA-expressing bacteria for the indicated genes in a systemic (A) and TRN-autonomous (B) RNAi strain. $^{**}P < 0.005$, $^{*}P < 0.05$; ns. The underlying numerical data and statistical analysis for each figure panel can be found in S1 and S2 Datasets, respectively. AVM, anterior ventral microtubule; dsRNA, double-stranded RNA; GABA, γ-aminobutyric acid; *gad*, glutamate decarboxylase enzyme gene; ns, not significant; RNAi, RNA interference; TRN, touch receptor neuron.

the neuronal integrity of ALM, PLM, and AVM neurons in a time-dependent fashion. ALM and PLM protection in *E. coli* HT115 was not further increased by the down-regulation of DAF-2 (Fig 10A and 10B, neuronal integrity on *E. coli* OP50 of ALM, PLM, and AVM at 25 °C in S11A and S11C Fig), suggesting that HT115-mediated protection involves the down-

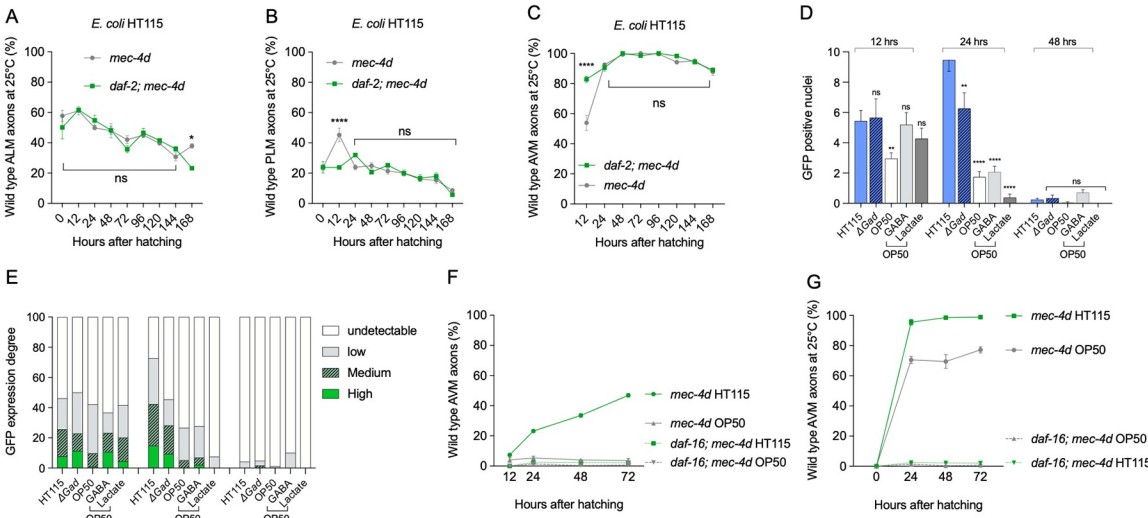

**Fig 10. Effect of down-regulation of the insulin pathway on *E. coli* HT115–mediated neuroprotection.** (A–C) Neuronal integrity of ALM (A), PLM (B), and AVM (C) neurons of *daf-2(ts)*; *mec-4d* animals feeding on HT115 food. (D) Number of GFP-positive nuclei in DAF-16::GFP animals feeding on *E. coli* OP50 or HT115 during development. DAF-16::GFP is observed in the entire body. (E) Degree of GFP expression of animals in (D). (F and G) Time course of neuronal degeneration in *daf-16; mec-4d* animals at 20 °C (E) and 25 °C (F). $^{****}P < 0.0001$, $^{**}P < 0.005$; ns. The underlying numerical data and statistical analysis for each figure panel can be found in S1 and S2 Datasets, respectively. ALM, anterior lateral microtubule; AVM, anterior ventral microtubule; DAF-2, ortholog of the insulin receptor; DAF-16, ortholog of the FOXO transcriptional factor; GABA, γ-aminobutyric acid; *gad*, glutamate decarboxylase enzyme gene; GFP, green fluorescent protein; *mec-4d*, mechanosensory abnormality protein 4; ns, not significant; PLM, posterior lateral microtubule.

regulation of the DAF-2 pathway. Owing to the cumulative protection effects of temperature and diet, *mec-4d* AVM neurons at 25 ˚C on HT115 diet reached almost 100% of wild-type axons, and the *daf-2* mutation maintained maximum protection (Fig 10C).

Because DAF-2 down-regulation causes the translocation of DAF-16 (ortholog of the FOXO family of transcription factors) to nuclei [31], we tested whether HT115 diet promoted nuclear translocation of GFP in a DAF-16::GFP-expressing strain (CF1139 strain, see Materials and methods). We fed CF1139 animals with *E. coli* OP50 and HT115 and compared the number of fluorescent nuclei in the entire body of animals at 12, 24, and 48 hours after hatching on each diet. Additionally, we qualified the intensity of GFP expression (see Materials and methods). *E. coli* HT115 promoted a significantly higher translocation of DAF-16 compared with *E. coli* OP50 at 24 hours only, returning to basal levels at 48 hours (Fig 10D). Given that bacterial GAD enzyme and GABA are correlated with the protection conferred by HT115 (Fig 8C), we tested whether they were involved in DAF-16 nuclear translocation. The Δ*gad* mutation caused a significant decrease in both the number of GFP-positive nuclei and the intensity of GFP expression (Fig 10D and 10E). This decrease, however, did not entirely eliminate DAF-16 nuclear expression, suggesting that in the context of *E. coli* HT115, GAD enzyme is not the only factor responsible for activation of DAF-16. The addition of GABA to *E. coli* OP50, although effective in conferring neuroprotection, failed to cause nuclear translocation of DAF-16. Similarly, lactate supplementation of *E. coli* OP50, though strongly protective of *C. elegans* AVM axons, also failed to promote DAF-16 translocation. This suggests that protection by bacterial metabolites GABA and lactate is likely caused by a mechanism independent of further translocation of DAF-16 to nuclei.

Next, we directly assessed the involvement of DAF-16 in the neuroprotective effect of *E. coli* HT115 by scoring neuronal integrity of the TRNs in *daf-16; mec-4d* animals feeding with HT115 at 20 ˚C and 25 ˚C. Both in *E. coli* OP50 and HT115, most TRN types were absent at birth in either temperature, with a marginal presence of AVMs, which also rapidly underwent degeneration in the *daf-16* mutant (Fig 10F and 10G, all categories in *E. coli* OP50 and HT115 at 20 ˚C and 25 ˚C in S11D and S11G Fig). This demonstrated that DAF-16 is required for the neuroprotection effect of the HT115 diet. Notably, the *daf-16* mutation completely abolished the protection of TRNs previously observed at 25 ˚C in *mec-4d* background [22] in both bacteria, suggesting that the *daf-16* mutation significantly lowers the threshold for neurodegenerative stimuli. The effect of increased degeneration in *daf-16; mec-4d*, though much more dramatic in *E. coli* HT115, is also observed in OP50. We assessed the effect of *daf-16* mutations in normal development and integrity of the TRNs in the strain WCH40 (*daf-16*[*m27*]; *uIs31* [$P_{mec-17}$*mec-17::gfp*]) expressing GFP in all the TRNs. DAF-16 loss alone did not cause an observable effect in the morphology of the TRNs (S11H Fig). Taken together, these results show that the function of DAF-16 is required for *E. coli* HT115–mediated protection to take place.

## Discussion

The relationship between bacteria and host affects virtually every studied aspect of an animal's physiology. However, whether bacteria and their metabolites can modulate neuronal degeneration is not known. In this work we show that bacterial diet dramatically influences neuronal outcomes in a *C. elegans* model of neurotoxic death triggered by the MEC-4d degenerin. *E. coli* HT115, a derivative of the K-12 strain, is the most protective of all bacteria tested, which included nonpathogenic laboratory bacteria, mild pathogens, and natural commensal bacteria. We found that this bacterium protects embryonic and postembryonic TRNs as well as the PVC interneuron, suggesting a pleiotropic effect on the nervous system. By comparing the

genomes, transcriptomes, and metabolomes of the most and least protective *E. coli* strains, we found that GABA is a key bacterial neuroprotective metabolite. Systemic *C. elegans* GABA receptors GAB-1 and LGC-37 and GABA transporter UNC-47 are required for wild-type neuroprotection conferred by HT115 bacteria. In addition to GABA, bacterial lactate was able to confer large neuroprotection. Importantly, HT115 neuroprotective effect can only take place on the condition of a functional DAF-16 transcription factor.

## Gut microbes regulate neurodegeneration in *C. elegans*

Recent work has highlighted the importance of the gut microbiota in shaping human health and well-being. Not only do intestinal microbes regulate many aspects of host physiology and development, but they have also been linked to mood disorders and neurodegenerative diseases [7,32,33]. For example, patients with Parkinson disease (PD) present constipation as a nonmotor symptom in early stages of the disease, which correlates with dysbiosis of the gut microbiome. Although it is unclear whether alterations in the gut microbiome are causes or consequences of these illnesses in the nervous system, emerging evidence using fecal transplantation in animal models demonstrates the ability of healthy and diseased gut microbiomes to ameliorate symptoms and confer disease, respectively [34,35].

Intestinal microbiota regulates central nervous system (CNS)-related traits through the microbiota–gut–brain axis. This consists of a bidirectional communication between the microbiota in the intestinal tract and the brain through the production of neuroactive molecules. Few recent studies correlate bacterial taxa from mammal's microbiota with metabolite production and its impact on brain function and pathology [36].

The bacterivore nematode *C. elegans* offers an advantaged platform for dissecting specific neuroprotective metabolites because bacteria can be given monoaxenically to animals that express a genetically encoded degenerative trigger (*mec-4d*). Dietary bacteria are usually broken up in the grinder located in the worm's pharynx right after ingestion [37]. This releases bacteria contents into the gut of the worm, a process that may also occur by explosive lysis [38,39] Nonetheless, a number of dietary bacteria survive this initial interaction to live and colonize the worm's digestive tract [40]. We show that UV-killed *E. coli* HT115 is equally as neuroprotective as live bacteria, ruling out that bacteria need to colonize the intestine and actively trigger a host immunoprotective response. Moreover, HT115 fed only for a few initial hours to newly hatched animals was sufficient to provide protection to adult neurons. This suggests that HT115 metabolites turn on a signaling cascade that outlasts the presence of the protective bacterial metabolite itself.

## Bacterial GABA is neuroprotective

The GAD system is a key mechanism used by intestinal bacteria to cope with acidic stress. This enzyme that produces GABA from glutamic acid is uniquely expressed in *E. coli* HT115. Genes coding for GAD are conserved in the genome of *E. coli* OP50; nonetheless, this strain lacks *rcsB*, encoding a transcriptional factor known to be required for the expression of *gad* genes in *E. coli*. A close inspection to the genome of OP50 reveals that this strain lost a chromosomal segment spanning from *micF* up to *yfaA*, which includes *rcsBCD* genes. This region is substituted by an IS1 in OP50, which could account for the low production of GABA in this strain. Notably, it has been reported that growth of an *E. coli* K-12 strain in buffered acidic conditions promotes the loss of several systems employed to grow in unbuffered acidic conditions, including GABA production [41]. OP50 is a derivative of *E. coli* B2. Like OP50, the B2 strain has a diminished GABA production and also lacks the *rcs* region. Thus, this and other traits may resemble the different selective pressures that have been present over the *E. coli* K-

12 and B lineages. Nonetheless, HT115 has a highly increased capacity to produce GABA, even compared with other K-12 strains. In fact, even among different unrelated bacteria tested, only *P. aeruginosa* PAO1 seemed to produce similar levels of GABA (Fig 8B). The origin of GABA overproduction in HT115 is not evident from genomic analyses and requires further research.

GABA can be metabolized to succinic semialdehyde (SSA) and succinate or translocated to the periplasm by the glutamate/GABA antiporter (GADT, [42]). GADT is dramatically overexpressed in protective bacteria, likely tilting the balance toward the accumulation of GABA in the periplasm. Unbiased metabolomic analysis identified GABA as one of the main metabolites differentially expressed in *E. coli* HT115 compared with OP50. Finally, GABA supplementation to *E. coli* OP50 is sufficient to provide neuroprotection.

Some human enteric microbiota members have been shown to require GABA to grow, and thus the production of GABA by enteric members of the *Bacteroides*, *Parabacteroides*, and *Escherichia* genera has been suggested to delineate the composition of the human microbiota [9]. Moreover, in the same work, it was shown that the relative abundance of GABA-producing members in intestinal microbiota negatively correlates with depression-associated brain signatures in patients. Bacterial GABA was proposed as one of the main effectors of microbiota on the CNS [8,9]. How bacterially produced GABA in the gastrointestinal tract may affect the brain or other distal CNS traits is intriguing. In animals, GABA receptors are expressed in epithelial cells, and a limited capacity of GABA to cross the blood–brain barrier has been reported [43]. However, it is not clear whether GABA and other microbiota-produced neuroactive molecules may exert their effect by localized or systemic activation of signaling pathways. In *C. elegans*, the number and type of GABA-containing cells and cells expressing GABA-uptake proteins and receptors are higher than previously thought and include nonneuronal cells [44]. Thus, although no intestinal GABAergic cells were reported, GABA could be pleiotropically sensed by a number of cells. In an infection model, *Staphylococcus aureus* molecules can trigger neuroendocrine reactions in *C. elegans* [45]. In our work, by using systemic and TRN-specific reverse genetics, we show that systemically delivered dsRNA of *unc-47*, *lgc-37*, and *gab-1* decreases neuroprotection in HT115 bacteria. This shows that GABA can be processed in nonneuronal cells to promote neuroprotection.

## DAF-16/Forkhead box O transcription factor loss impairs *E. coli* HT115 protection

A phosphorylation cascade downstream of the insulin receptor DAF-2/IGF1R controls DAF-16 transcription factor activation. When DAF-2/IGF1R is activated, DAF-16/Forkhead box O (FOXO) is prevented from entering the nuclei [31]. Starvation, pathogen exposure, and other interventions promote its nuclear translocation and activation of transcriptional targets [46]. DAF-16 can also be directly activated upon fungal infection in a DAF-2-independent fashion [47]. In our experiments, DAF-2 inactivation did not increase HT115 protection, suggesting they act in the same pathway. Neuroprotective and neuroregenerative effects of DAF-2 downregulation involve the function of DAF-16 [16,22,48]. Comparison of DAF-16 nuclear translocation between diets showed that at 24 hours DAF-16 was increased in the nuclei of animals feeding on *E. coli* HT115 compared with OP50. Nuclear localization of DAF-16 returned to basal levels at 48 hours. This raises the possibility that DAF-16 activity during that window of time is sufficient for long-term protection by means of the stability of its transcriptional targets. Strikingly, although a *daf-16* mutation did not cause TRN loss of integrity in wild-type animals, it completely abolished the protection by HT115 diet on *mec-4d* mutants. Taken together, these results show that DAF-16 is critical under conditions of neuronal stress and necessary for the dietary protection mediated by *E. coli* HT115 to take place.

One of the metabolites that are overabundant in HT115 compared with OP50 is GABA. Interestingly, loss of the GAD enzyme, which catalyzes GABA production, diminished DAF-16 translocation in HT115 bacteria. GABA alone, however, supplemented to *E. coli* OP50 did not promote nuclear localization of DAF-16. These results suggest that although GABA is required for DAF-16 localization, it is not sufficient by itself, and other metabolites produced by HT115, such as lactate, act cooperatively to produce this translocation. Alternatively, this could suggest that although DAF-16 expression is required for neuroprotection to occur, neuroprotective metabolites likely function through a mechanism that does not involve active translocation of DAF-16 above basal levels. Currently, we cannot discriminate between these two possibilities, and more research is needed to further clarify the molecular pathways involved in GABA-mediated effects in neuronal integrity preservation by HT115. This work reveals a complex scenario of communication between bacteria and host that is likely to involve systemic and neuron-specific changes.

## Materials and methods

### *C. elegans* maintenance and growth

Wild-type (N2) and mutant strains TU2773 ($uIs31[P_{mec-17}mec-17::gfp]$; $mec-4d[e1611]$), CF1139 ($daf-16[mu86]$; $muIs61[P_{daf-16}daf-16::gfp]$; $rol-6[su1006]$), WCH34 ($daf-2ts[e1368]$; $mec-4d[e1611]$; $uIs31 [P_{mec-17}mec-17::gfp]$), WCH39 ($daf-16[m27]$; $mec-4d [e1611]$; $uIs31[P_{mec-17}mec-17::gfp]$), WCH40 ($daf-16[m27]$; $uIs31[P_{mec-17}mec-17::gfp]$), TU38 ($deg-1[u38]$), TU3755 ($uIs58[P_{mec-4}mec-4::gfp]$), WCH6 ($uIs71[P_{mec-18}sid-1;Pmyo-2mcherry]$, $uIs31[Pmec-17mec-17::gfp]$, $sid-1[pk3321]$, $mec-4d[e1611]$), and WCH41 ($uIs58 [P_{mec-4}mec-4::gfp]$; $mec-4d[e1611]$) were grown at 20 ˚C as previously described [49]. All nematode strains are maintained on *E. coli* OP50 strain prior to feeding with other bacteria. Unless otherwise noted, all plates were incubated at 20 ˚C.

### Bacterial growth

Bacteria were grown overnight on Luria-Bertani (LB) plates at 37 ˚C from glycerol stocks. The next morning, a large amount of the bacterial lawn was inoculated in LB broth and grown for 6 hours on agitation at 450*g* at 37 ˚C. In all, 100 mL of this bacterial culture was seeded onto 60-mm NGM plates and allowed to dry overnight before worms were placed on them. We used the following bacterial strains as worm food: *E. coli* OP50, *E. coli* HT115, *E. coli* K-12 BW25113, *E. coli* B BL21 DE3, *C. aquatica*, *C. testosteroni*, *B. megaterium*, *P. aeruginosa* PAO1, *Pseudochrobactrum* sp., *Stenotrophomonas* sp., *B. pumilus*, and *E. coli* HT115 Δ*gad*.

**Pertinence of the strains selected.** We chose a diverse array of bacterial strains that have been used before in *C. elegans* research as well as bacteria from a wild microbiome. The rationale for using laboratory strains is (1) their availability and (2) further characterization of strains currently used and whose effect in neuroprotection and metabolite production may be helpful in result interpretation for the larger *C. elegans* research community. Specifically, we used strains for routine laboratory work that served as a control (*E. coli* OP50); strains for dsRNA delivery in *C. elegans* RNAi experiments (*E. coli* HT115); the two *E. coli* strains from which OP50 and HT115 were derived (*E. coli* B and K-12, respectively); a mild pathogen used to study bacteria–host interactions (*P. aeruginosa* PAO1); *C. aquatica*, a high-quality food for *C. elegans*; and *B. megaterium*, which causes extreme roaming and is hard to eat. Wild bacteria were included because they were isolated from wild *C. elegans* and constitute a natural microbiome.

## UV killing of bacteria

**Killing on plates.**   Bacterial cultures (300 μL) grown at an OD of 0.8 were inoculated on 60-mm NGM plates. Once the liquid dried (overnight), plates were placed upside down in the UV transilluminator (Cole-Palmer HP 312 nm) for 5 minutes at high power. Synchronized animals were seeded onto the plates immediately after bacterial killing. An inoculum of the UV-treated bacterial lawn was picked and streaked onto new LB plates to confirm that bacteria were effectively killed.

**Killing in liquid.**   Bacterial cultures (5 mL) grown at an OD of 0.8 were placed in an empty 60-mm plate and exposed to UV light for 10 minutes with slow agitation every 5 minutes. We used 300 μL of this liquid to seed 60-mm plates.

## Criteria for neuronal integrity

**AVM neuron.**   Morphological evaluation of AVM neuron was modified from Calixto and colleagues, 2012 [16]. Neurons with full-length axons, as well as those with anterior processes that passed the point of bifurcation to the nerve ring, were classified as AxW (see Fig 1A). Axons with a process connected to the nerve ring were classified as AxL, and those that did not reach the bifurcation to the nerve ring were classified as AxT. Lack of axon, only soma, and soma with only the ventral projection were classified as Axϕ.

**ALM and PLM neurons.**   Neurons with full-length axons were classified as AxW. AxT were ALM neurons with axons that did not reach the bifurcation to the nerve ring, and PLMs were axons that did not reach mid body. Neurons without axons or somas were classified as Axϕ.

## Microscopy

For morphological evaluation, worms were mounted on 2% agarose pads, paralyzed with 1 mM levamisole, and visualized under a Nikon Eclipse Ti-5 fluorescence microscope with 40× or 60× magnification under Nomarski optics or fluorescence. For high-resolution images, we used a Leica TCS SP5X microscope. DAF-16 nuclear expression and MEC-4 localization in CF1139 and TU3755 animals, respectively, were quantified using ImageJ (1.46v). For accuracy in the categorization and to avoid damage due to long exposure to levamisole, animals were scored within 20 minutes after placing them on the agarose pads.

## Feeding RNAi

Clones of interest were taken from the Ahringer RNAi bacterial library [50,51]. We performed a P0 screen as described in Calixto and colleagues [16] in *mec-4d* animals and TRN-autonomous RNAi strain carrying the *mec-4d*(*e1611*) mutation (WCH6, [16]). Bacterial clones were taken from glycerol stocks and grown overnight on LB plates containing tetracycline (12.5 μg/mL) and ampicillin (50 μg/mL). The next morning, a chunk of bacterial lawn was grown on liquid LB containing ampicillin (50 μg/mL) for 8 hours. A total of 400 μL of bacterial growth was plated on NGM plates containing 1 mM IPTG and carbenicillin (25 μg/mL) and allowed to dry until the next day, when 30–50 newly hatched *mec-4d* or WCH6 worms were placed on NGM plates. ALM integrity was scored in young adults 72 hours later.

## Synchronization of animals

Plates with large amounts of laid eggs were washed with M9 to eliminate all larvae and adults. Within the next 2 hours, newly hatched L1 animals were collected with a mouth pipette and transferred to the desired experimental plates.

## Time course of neuronal degeneration

Synchronized L1 larvae were placed in plates at 20 ˚C with the desired bacterial food using a mouth pipette. We scored the integrity of the AVM neuron axon (1) during development at 12, 24, 48, and 72 hours posthatching; (2) every 24 hours for longer periods (from 0 or 12 hours until 168 hours posthatching); and (3) at adulthood (72 hours) after each treatment. The same experiments at 25 ˚C also included ALM and PLM neurons. For experiments at 25 ˚C without temperature shifts, parents of animals examined were kept at the same temperature starting as L4s. Animals were scored until 168 hours at intervals of 24 hours. For each evaluation, we used at least three biological replicas with triplicates of 30 worms each.

## Food changes

Synchronized L1 animals were placed in NGM plates seeded with UV-killed *E. coli* HT115 or OP50 for 6 or 12 hours and later changed to *E. coli* OP50 or HT115 until 72 hours posthatching. To transfer 6- and 12-hour-old animals to the new bacteria, larvae were washed off from the plates with sterile water supplemented with carbenicillin (25 μg/mL). Each replica was collected using a pipette in an Eppendorf tube. Animals were subsequently centrifuged for 2 minutes at 450*g*. The pellet was washed with sterile water supplemented with carbenicillin followed by centrifugation. After two washes, the pellet was collected and transferred to new plates. Axonal morphology was evaluated at 12, 24, 48, and 72 hours. For controls in both experiments, we grew synchronized animals in the same way on *E. coli* OP50 and HT115 for their entire development until 72 hours.

## DAF-16 localization on different foods

The 30 L4 CF1139 (*daf-16*[*mu86*]; *muIs61*[$P_{daf-16}$*daf-16::gfp*]; *rol-6*[*su1006*]) animals were placed in plates seeded with *E. coli* OP50, wild-type HT115, HT115 Δ*gad* mutant, or *E. coli* supplemented with 2 mM lactate or 2 mM GABA and allowed to lay eggs for 24 hours. The 30 L1 larvae, synchronized 0–2 hours posthatching, were transferred with a mouth pipette to new plates with *E. coli* HT115 or OP50. Morphological evaluation was performed at 12, 24, and 48 hours posthatching.

We quantified GFP-positive nuclei in whole body. The most visible expression was in intestinal and hypodermal cells.

## MEC-4 puncta quantification at different temperatures

The 30 L4 TU3755 (*uIs58* [$P_{mec-4}$*mec-4::gfp*]) worms were fed *E. coli* OP50 or HT115 and allowed to lay embryos for 24 hours. The 30–50 L1 larvae were synchronized as described above and placed on the corresponding diet at 15, 20, or 25 ˚C.

## Criteria for DAF-16 nuclear localization

We counted the number of GFP-positive nuclei between the terminal pharyngeal bulb and 30 μm after the vulva (or developing vulva). Because we performed a time course evaluation, animals had different body sizes at each time point. The comparisons were made between animals of the same life stage feeding on the two bacteria. For 12-hour-old animals, pictures were taken in a 60× objective, and for 24- and 48-hour-old animals, pictures were taken in a 40× objective. Experiments were done in triplicate for each time point.

Intensity of nuclear expression was estimated as "high" if there were more than 20 GFP-positive nuclei, "medium" between 10 and 19, "low" between 3 and 9, and "undetectable" if there were fewer than 3.

## MEC-4 puncta quantification

One PLM per L4 animal was photographed under a 40× objective. The number of puncta was counted in 100 μm of axon starting from the neuronal soma. Images were visualized in ImageJ (1.46v), and puncta were counted manually.

## Touch response

To evaluate the functionality of the AVM mechanoreceptor neuron and PVC interneuron, the ability of animals to respond to gentle touch was tested. Animals were touched at 20 ˚C.

First, animals were synchronized in L1 larvae and placed 30 per NGM plate seeded with different bacteria (mentioned above). Then, in the case of AVM neuron, animals were touched with an eyebrow one time in the head, gently stroking where the pharyngeal bulb lies at 72 hours, whereas for the PVC interneuron, animals were gently touched with an eyebrow 10 times in a head-to-tail fashion every 24 hours after hatching.

## Temperature shifts

To evaluate the effects of DAF-2 down-regulation on dauer animals, we used the strains WCH34 (*daf-2ts*[*e1368*]; *uIs31* [$P_{mec-17}$*mec-17::gfp*]; *mec-4d*[*e1611*]) and TU2773 (*uIs31*[$P_{mec-17}$*mec-17::gfp*];*mec-4d*[*e1611*]) as a control. L4 animals from both strains were placed on plates seeded with *E. coli* HT115 and OP50 at 25 and 20 ˚C. Then, animals were synchronized by taking 30–50 L1 larvae and placing them in three plates for each point of evaluation. Morphology of TRNs (AVM, PLMs, ALMs, and PVM) was scored from 12 to 168 hours with intervals of 24 hours.

## Longitudinal analysis of neuronal degeneration

At hatching (0–2 hours), the 30 synchronized L1s were placed on individual plates seeded with *E. coli* HT115. Animals were examined every 24 hours for 3 days, starting at 24 hours post-hatching, by placing them on 2 μl of 0.1-μm polystyrene beads for AVM observation and photography. We used polystyrene beads in order to maintain the shape of the animal, allowing its rescue from the agarose pad for posterior visualizations. Each animal was gently returned in M9 to the plate with a mouth pipette.

## Bacterial growth with controlled OD

*E. coli* HT115 bacteria were inoculated in LB starting from a −80 ˚C glycerol stock and allowed to grow for 1 hour. Nine different falcon tubes were used to grow bacteria. After the first hour, 1 mL from each tube was taken to measure the OD using a spectrophotometer (Ultraspec 2100). When the cultures reached the desired OD, growth was stopped. This procedure was repeated for each measurement. To obtain the lowest values (0.4/0.6/0.8), cultures were evaluated every hour and, for the rest of the values (0.8–2.0), every 15 or 30 minutes. A total of 200 μL of each culture was inoculated onto six NGM plates. Seeded plates were dried on a laminar flow hood for at least 1 hour. Finally, bacteria were killed on a UV transilluminator (Cole-Palmer High performance) by placing the open plate upside down for 5 minutes using the highest wavelength (365 nm). On three of those plates, 30 synchronized L1 worms were placed. The other three plates were used to test whether bacteria were killed during the protocol. To this end, parts of the lawn of UV-killed bacteria were streaked onto another LB plate to observe whether bacteria grew on the agar within the next 3 days.

## Supplementation with bacterial supernatant

Overnight *E. coli* OP50 and HT115 bacterial cultures (5 mL) were centrifuged for 15 minutes at 3,500*g*. The supernatant was sieved twice on 0.2-μm filters using a sterile syringe to separate bacteria in suspension. *E. coli* OP50 supernatant was used to resuspend *E. coli* HT115 pellet, and the same was done with *E. coli* HT115 supernatant on OP50 pellet. Each bacterial suspension was inoculated on NGM plates as described above.

## Supplementation with GABA, glutamic acid, or lactate

After liquid UV killing, the desired bacteria were mixed with 2 mM GABA, 2 mM glutamate, or 2 mM of L-lactate (Sigma). Bacterial cultures (400 μL) were inoculated in 60-mm NGM plates to cover the entire surface. The next day, 0- to 2-hour synchronized L1 worms were placed in the NGM plates for 24 hours. Animals were moved to freshly prepared plates every 24 hours for 72 hours.

## Mix of bacteria and UV killing

Bacterial cultures were combined in different proportions with the other bacteria or LB for controls. Liquid LB media growth until OD 0.6 of *E. coli* OP50 or HT115 were diluted 1:125 in LB and incubated at 37 ˚C with shaking O/N. Bacterial cultures were mixed in the indicated proportions and used to prepare plates as described before (0.1%, 1%, 10%, and 50% of *E. coli* HT115 in OP50). Plates with mixed bacteria were irradiated with UV before adding the worms.

## Generation of bacterial *gad* mutant

Two orthologs of the *gad* gene are present in *E. coli* HT115, denominated *gadA* and *gadB*. *E. coli* HT115 mutants were constructed using homologous recombination with PCR products as previously reported [52]. Wild-type *E. coli* HT115 [53] was transformed with plasmid pKD46 [52]. Next, electrocompetent bacteria were prepared at 30 ˚C in SOB medium with ampicillin (100 μg/mL) and arabinose and electroporated with a PCR product obtained using the set of primers gadBH1P1 and gadBH2P2 and pKD3 as template. Recombinant candidates were selected in LB plates plus chloramphenicol at 37 ˚C. Colonies were tested for loss of ampicillin resistance. Amp^s colonies were checked for substitution of the *gadB* gene by the chloramphenicol acetyl transferase cassette by PCR using primers gadB-A and gadB-B, which flank the insertion site. This rendered the *E. coli* HT115 Δ*gadB*::*cat* derivative.

This strain was electroporated with pKD46. Competent cells of this transformed strain were prepared with ampicillin and arabinose and electroporated with a PCR product generated using primers NgadAH1P1 and NgadAH2P2 and pKD4 as template.

Recombinant candidates were selected in LB plates plus kanamycin (30 μg/mL) at 37 ˚C. Colonies obtained were tested for loss of ampicillin resistance and checked for substitution of the *gadA* gene by the kanamycin resistance gene through PCR using primers NGadAFw and NgadARv flanking the substitution site. This yielded an *E. coli* HT115 Δ*gadB*::*cat*/Δ*gadA*::*kan* double mutant strain (referred as *E. coli* HT115Δ*gad* mutant in the text). All primers are listed in S3 Table.

## Generation of the pG*gadA* plasmid

*E. coli* HT115 *gadA* gene was amplified by PCR using primers NgadAFw and NgadARw with Taq polymerase and cloned in the pGemT Easy plasmid (Promega) according to the manufacturer's instruction to generate pG*gadA*. *E. coli* OP50 and *E. coli* HT115 Δ*gadB*::*cat*/Δ*gadA*::*kan*

were transformed chemically with the pG*gadA* plasmid. Competent bacteria were obtained using a calcium chloride protocol [54]. Transformed colonies for *E. coli* OP50 + pG*gadA* and HT115 Δ*gadB*::*cat*/Δ*gadA*::kan+pG*gadA* were confirmed by antibiotic resistance (ampicillin 100 μg/mL), plasmid purification, plasmid length, and gene endonuclease restriction with XbaI (NEB).

## GAD enzymatic activity

Glutamate decarboxylase enzymatic activity was measured according to Rice and colleagues [29], with modifications [28]. Fresh bacterial colonies were grown in LB with the appropriate antibiotic until 108 CFU/mL (3–4 hours). Antibiotics used were 25 μg/mL of streptomycin for *E. coli* OP50, 25 μg/mL of tetracycline for HT115, 20 μg/mL of kanamycin and chloramphenicol for HT115Δ*gad*, and 25 μg/mL of streptomycin and ampicillin and 25 μg/mL of IPTG for OP50 + pG*gadA*.

Then, 10 mL of each culture was centrifuged at 500*g* for 10 minutes, and the pellet was resuspended in 5 mL of phosphate buffer (KPO$_4$ 1M [pH 6.0]). This step was repeated one more time, and the pellet was resuspended in 2 mL of GAD reagent (1 g glutamic acid [Sigma], 3 mL Triton X-100 [Winkler], 0.05 g bromocresol blue [Winkler], 90 g sodium chloride [Winkler] for 1 L of distilled water with final pH 3.4) and preserved at 4 ˚C for 2 months maximum. Once the pellet was resuspended, the samples were measured for colorimetric differences (UltraSpect 2100) at 620 nm. Liquid LB treated without bacteria is used as a blank. Average colorimetric value for *E. coli* HT115 was considered 100% of possible GAD activity for each replicate.

## Growth of bacterial cultures for GABA quantification by GABase assay

Bacterial cultures were grown in LB media with 25 μg/mL of streptomycin for *E. coli* OP50, 25 μg/mL of tetracycline for HT115, and 20 μg/mL of kanamycin and chloramphenicol for HT115Δ*gad* until $10^7$ CFU/mL (3–4 hours).

## Bacterial GABA quantification by GABase assay

This reaction consisted of two enzymes that convert a molecule of GABA to succinate by GABA transaminase (GAT) and SSA dehydrogenase (SADH), producing detectable NADPH at 340 nm. Additionally, by inhibition of GAT with aminoethyl hydrogen sulfate, substrate concentrations for each reaction are distinguishable according to OD [NADPH] total = OD [NADPH] GAT + OD [NADPH] SADH.

**Bacterial GABA quantification.** Measurement of GABA was performed following the GABase Sigma protocol with modifications [30].

Reaction was performed in 100-μl final volume containing 1U/mL of GABase previously mixed with 75 mM of potassium phosphate buffer in 25% of glycerol, 100 mM of pyrophosphate potassium buffer, 500 μM of NADP+, 5 mM of a-ketoglutarate, 100 μM of dithiothreitol, 50 mM of aminoethyl sulfate as GAT inhibitor, and 10 uL of each sample. All reactions were performed in 96-well Nunclon plates in a NanoQuant 200 Infinite spectrophotometer for 1 hour at 340 nm and 37 ˚C. Every run contained positive (5 mM of GABA) and negative controls (no GABAse or no GABA) to be subtracted as background of total NADPH measurement, as well as inhibited reaction with respective controls. The GABA concentration curve was performed in triplicate to measure absorbance from 0 to 5 mM of GABA (Abs = 0.1138 × [GABA] + 0.04522).

**GABA extraction for GABase reaction.** All bacteria culture extractions were performed from $1 \times 10^8$ CFU/mL three times in each triplicate. A total of 1 mL of culture was centrifuged

at 1,100*g* (4,000 rpm) for 10 minutes, and pellet was washed twice with phosphate buffer (1 M KPO$_4$ [pH 6.0]) by pipetting and centrifugation at 3,300*g* (7,000 rpm) for 10 minutes. After the last centrifugation, 100 μL of deionized water was added, and samples were treated in a water bath at 95 ˚C for 15 minutes. Then, a final centrifugation at 1,100*g* (4,000 rpm) for 10 minutes separated GABA in the supernatant from bacteria debris. Extracted samples were frozen at −20 ˚C until immediate use in the reaction the next day.

**Correlation analysis.** Neuronal morphology for *mec-4d* worms, GAD activity, and GABA production in each bacterium were correlated using one-tailed Pearson's correlation analysis, and linear regression confirmed the variables' slope differences.

## Growth of bacteria for NMR spectroscopy

Bacterial strains were grown in solid LB media with 25 μg/mL of streptomycin for *E. coli* OP50, 25 μg/mL of tetracycline for *E. coli* HT115, and 20 μg/mL of kanamycin and chloramphenicol for HT115Δ*gad* at 37 ˚C for 10 hours. Eight preinocula of 2 mL of liquid LB were set for each bacterial strain, grown overnight, and continued in 35 mL of LB and monitored by OD until required (OD of 1).

## Metabolite extraction for NMR

Bacterial cultures were centrifuged at 4,000*g* for 5 minutes to separate bacteria from the media. Bacterial pellet was resuspended in phosphate-buffered saline (PBS: NaCl [137 mM], KCl [2.7 mM], Na$_2$HPO$_4$ [10 mM], and KH$_2$PO$_4$ [1.8 mM]) and washed twice after centrifugation with PBS at 4,000*g* for 5 minutes. After the last wash, each pellet was resuspended in 1 mL of cold extraction buffer and pipetted into Eppendorf tubes. The extract solution was prepared by mixing equal volumes of acetonitrile and KH$_2$PO$_4$/NaH$_2$PO$_4$ (100 mM, pH 7.4).

Bacterial membrane permeabilization was performed in two steps. First, Eppendorf tubes were submerged in a liquid nitrogen bath for 2 minutes, defrosted at 4 ˚C, and vortexed for 30 seconds. This procedure was repeated three times. Secondly, tubes from the first step were sonicated in an ultrasonic water bath (Bioruptor UCD-200, Diagenode) for 15 cycles of 30 seconds on and 30 seconds off at full power. Finally, to obtain the metabolites, samples were centrifuged for 10 minutes at 8,000*g*, and the supernatant was recovered. This step was repeated to optimize metabolite recovery. Samples were dried in a vacuum dryer (Savant) for 60 minutes at 50 ˚C and 300*g*.

**Metabolic profiling.** The 1H NMR spectroscopy and multivariate data analysis were performed at Plataforma Argentina de Biología Estructural y Metabolómica (PLABEM).

## Sample preparation for 1H NMR spectroscopy

Samples were randomized and reconstituted in 600 μL of 100 mM Na$^+$/K$^+$ buffer (pH 7.4) containing 0.005% TSP (sodium 3-trimethylsilyl- (2,2,3,3-2H4)-1-propionate) and 10% D$_2$O. In order to remove any precipitate, samples were centrifugated for 10 minutes at 14,300*g* at 4 ˚C. The 500 μL of the centrifuged solution was transferred into a 5-mm NMR tube (Wilmad LabGlass).

## 1H NMR spectroscopic analysis of bacterial extracts

NMR spectra were obtained at 300 K using a Bruker Avance III 700-MHz NMR spectrometer (Bruker Biospin, Rheinstetten, Germany) equipped with a 5-mm TXI probe. One-dimensional 1H NMR spectra of bacterial extracts were acquired using a standard 1-D noesy pulse sequence (noesygppr1d) with water presaturation [55,56]. The mixing time was set to 10 milliseconds,

the data acquisition period to 2.228 seconds, and the relaxation delay to 4 seconds. The 1 H NMR spectra were acquired using four dummy scans and 32 scans with 64 K time domain points and a spectral window of 20 ppm. FIDs were multiplied by an exponential weighting function corresponding to a line broadening of 0.3 Hz. Two-dimensional NMR spectra 1H-1H TOCSY and 1H J-resolved pulse sequences were acquired for resonance assignment purposes.

## Quality controls

Quality controls (QC) were prepared as suggested by Dona and colleagues [55]. The 1H NMR spectra of QC samples were acquired every eight study samples.

## 1H NMR spectral processing

Spectroscopic data were processed in Matlab (version R2015b, The MathWorks). Spectra were referenced to TSP at 0.0 ppm, and baseline correction and phasing of the spectra were achieved using Imperial College written functions (provided by T. Ebbels and H. Keun, Imperial College London). Each spectrum was reduced to a series of integrated regions of equal width (0.04 ppm, standard bucket width). Noninformative spectral regions containing no metabolite signals, TSP signal, and the interval containing the water signal (between 4.9 and 4.6 ppm) were excluded. Each spectrum was then normalized by the probabilistic quotient method [57]. Spectra alignment was made using the alignment algorithm recursive segment-wise peak alignment [58] in user-defined windows.

## Statistical analysis of NMR spectroscopic data

The preprocessed 1H NMR spectral data were imported to SIMCA (version 14.1, Umetrics AB, Umeå, Sweden) for multivariate data analysis. PCA was performed on the Pareto-scaled NMR dataset. OPLS-DA was made to maximize the separation between bacterial groups as a function of neuroprotection. To ensure valid and reliable OPLS-DA models and to avoid over-fitting, 200 permutations were carried out. Discriminant features between classes in OPLS-DA models were defined using a combination of loading plot (S-line plot) and VIP plot. Variables met highest p(corr)1 in S-Line plot, and VIP values >1 were selected and validated by spectral raw data examination.

## NMR resonances assignment

Two-dimensional NMR spectra 1H J-resolved pulse sequences were acquired for resonance assignment purposes. Discriminant features were assigned searching in the *E. coli* Metabolome Database (ECMDB) [59,60]. The unequivocal identification of GABA and glutamate were made through spike-in experiments over an *E. coli* HT115 and HT115 Δ*gad* extract, respectively.

## Growth of bacteria expressing *gad* plasmid

Genetically complemented bacteria with pG*gadA* were grown until an OD of 0.6. GAD expression was induced by adding 0.15 mM IPTG to OD-0.6 liquid cultures. After, 1-hour samples of each condition were taken to assess GAD activity and to seed on NGM plates with the appropriate antibiotic and supplemented with 0.1 mM of IPTG.

## Genome sequencing of *E. coli* OP50 and HT115

*E. coli* OP50 and HT115 were grown from glycerol stock on LB plates overnight. The next morning, portions of the lawn were cultured on agitation for 4 hours in liquid LB. Liquid cultures (2 mL) were pelleted, and DNA was purified using the UltraClean microbial DNA isolation kit (MO BIO) according to the manufacturer's instructions.

Whole-genome sequencing was done at Genome Mayor Sequencing Services. Paired-end reads ($2 \times 250$ bp) were generated using the Illumina MiSeq platform. Sequence data were trimmed using Trimmomatic version 0.27 [61]. Trimmed reads were assembled using SPAdes version 3.1.0 [62]. Genome annotation for both organisms was done using PROKKA version 1.9 [63]. Both genomes were deposited to NCBI under the accession numbers PRJNA526029 (*E. coli* OP50) and PRJNA526261 (*E. coli* HT115).

## Comparative genome analysis of *E. coli* OP50 and HT115

**Identification of common and unique sequences of E. coli strains.**   Annotated proteins in fasta format from assembled genomes of *E. coli* OP50 (PRJNA526029) and HT115 (PRJNA526261) were compared between strains by Reciprocal Best Hit analysis using Blast+ [64,65]. We ran Blastp [66] of the protein sequences of OP50 strain against HT115. Then, we ran Blastp of sequences from the *E. coli* HT115 strain against OP50. We extracted top hits based on bit scores and E-values. Then, we compared both top hits and selected the best match of each other, named "common sequences." All the other sequences were defined as "unique."

## Transcriptomic analysis

For total RNA isolation, *E. coli* OP50 and HT115 were grown from glycerol stock on LB plates overnight. The next morning, portions of the lawn were cultured on agitation in liquid LB for 5 hours. Then, bacterial cultures (2 mL) were pelleted for RNA extraction with Max Bacterial Trizol kit (Invitrogen) according to the manufacturer's protocol.

cDNA libraries for Illumina sequencing were generated by Centro de Genómica y Bioinformática, Universidad Mayor, Chile. cDNA libraries were made with Illumina Truseq stranded mRNA kit according to the manufacturer's protocol in Illumina HiSeq platform. QC of libraries was made with bioanalyzer, and quantification was done with qPCR StepOnePlus Applied Biosystem. Six sets of lllumina paired-end reads in FASTQ format corresponding to three replicates from *E. coli* strain HT115 and three replicates from strain OP50 were analyzed as follows.

**Data preprocessing and QC.**   Reads with an average quality lower than 30 over four bases, as well as reads shorter than 16 bp, were discarded with Trimmomatic version 0.35 [61]. Pre- and posttrimming quality visualization was made with FastQC (https://www.bioinformatics.babraham.ac.uk/projects/fastqc/).

**Mapping and quantification of transcript abundance.**   Mapping and quantification from *E. coli* HT115 and OP50 strains were made using Bowtie2 [67] with default parameters. We used as reference the ASM435494v1 assembly from *E. coli* HT115 (accession GCA_004354945.1) and ASM435501v1 from *E. coli* OP50 (GCA_004355015.1).

**Differential expression analysis.**   This analysis was performed preserving only common sequences; transcript abundance was compared between strains. For this, read quantification was performed with FeatureCounts in Rsubread [68] in R (version 3.5), and then differential expression analysis was performed using DeSeq2 version 3.8 [69] using default parameters. Cutoff for differentially expressed genes (DEG) was set at adjusted *p*-value (padj) $< 0.01$ and is reported in S2 File.

**Criteria for gene expression level.** We categorized gene expression according to [70] as follows: low if expression is between 0.5 and 10 FPKM or 0.5 and 10 TPM; medium if expression is between 11 and 1,000 FPKM or 11 and 1,000 TPM; and high if expression was more than 1,000 FPKM or 1,000 TPM.

## Biological and technical replicates

Each experiment was performed in three technical triplicates and at least three biological replicates. We define biological replicates as experiments made on different days, containing triplicates of each condition, and a technical replicate as a triplicate of the same condition on the same day. The average of the three reads of each triplicate is considered as one count. Each experiment has three technical replicates that were in turn averaged to constitute one of the points of each figure. Data are collected and processed as a single technical replicate (the average of three counts of the same plate), and its mean is used as a single biological replicate. Each figure contains at least three experiments (biological replicates) performed as explained before. All the biological replicates are performed spaced from each other from 1 day to 1 week.

**Sample size.** Each experiment started with at least 30 synchronized worms on each technical triplicate with the exception of the longitudinal study of degeneration on *E. coli* HT115, which used 30 animals in total.

## Statistical evaluation

Statistical evaluation was performed using one- or two-way ANOVA with post hoc tests and a Student t test when indicated. Results of all tests are detailed in S2 Dataset.

## Supporting information

**S1 Fig. Axonal categories at different optical density.** (A and B) All axonal categories (A) and wild-type axons (B) in worms feeding on *E. coli* HT115 bacteria grown to different optical density. The underlying numerical data and statistical analysis for each figure panel can be found in S1 and S2 Datasets, respectively.
(TIFF)

**S2 Fig. Axonal morphological categories of different classes of neurons in *E. coli* HT115 and OP50.** All axonal categories of animals feeding on *E. coli* OP50 (A, C, and E) and HT115 (B, D, and F) at 25 ˚C. The underlying numerical data and statistical analysis for each figure panel can be found in S1 and S2 Datasets, respectively.
(TIFF)

**S3 Fig. Quantification of mechanosensory channel MEC-4 expression *in vivo*.** Number of puncta in 100 μm of PLM axons on each bacterial diet (A) and representative photograph of PLM axons used for quantification. Size bar is 20 μm. The underlying numerical data and statistical analysis for each figure panel can be found in S1 and S2 Datasets, respectively. MEC-4, mechanosensory ion channel subunit; PLM, posterior lateral microtubule.
(TIFF)

**S4 Fig. Complete axonal categories of priming experiments.** (A–D) All axonal categories of animals feeding *E. coli* HT115 for 6 (A) and 12 (B) hours with controls of ad libitum *E. coli* OP50 (C) and HT115 (D) or feeding *E. coli* OP50 for 6 (E) and 12 (F) hours with controls of ad libitum *E. coli* OP50 (G) and HT115 (H). The underlying numerical data and statistical analysis for each figure panel can be found in S1 and S2 Datasets, respectively.
(TIFF)

**S5 Fig. Complete axonal categories for animals feeding on modified bacteria.** (A and B) All axonal categories of animals feeding wild-type and Δ*gad E. coli* HT115 (A) and OP50 (B) modified with Gad-expressing plasmids, glutamate, and GABA. The underlying numerical data and statistical analysis for each figure panel can be found in S1 and S2 Datasets, respectively. GABA, γ-aminobutyric acid; *gad*, glutamate decarboxylase enzyme gene.
(TIFF)

**S6 Fig. Multivariate analysis and model's validation.** (A) PC score plot derived from 1H NMR spectra indicating metabolic differences between wild-type *E. coli* strains OP50 (orange) and HT115 (green) and HT115 Δ*gad* mutant (light gray). Quality controls are displayed in yellow. Model parameters are R2X = 0.787 and Q2 = 0.705. (B and C) OPLS-DA validation by 200 permutations. *E. coli* HT115 and HT115 Δ*gad* validate model intercepts: R2 = (0.0; 0.656) and Q2 = (0.0; −0.626) (B). Protective *E. coli* HT115 and nonprotective strains *E. coli* OP50 and HT115 Δ*gad* validate model intercepts: R2 = (0.0; 0.329) and Q2 = (0.0; −0.566) (C). The underlying numerical data for each figure panel can be found in S1 Dataset. *gad*, glutamate decarboxylase enzyme gene; NMR, nuclear magnetic resonance; OPLS-DA, orthogonal projections to latent structures discriminant analysis; PC, principal component.
(TIFF)

**S7 Fig. Resonances assignment confirmation by NMR.** (A and B) 1H NMR spectra of GABA (A) and glutamate (C). (B–D) Spike-in of GABA and glutamate confirms identity of metabolites. *E. coli* HT115 extract (red) (B); *E. coli* HT115 Δ*gad* extract (blue). Spike-in was made adding 5 μL of standard 10 mM twice. The underlying numerical data and statistical analysis for each figure panel can be found in S1 and S2 Tables. GABA, γ-aminobutyric acid; *gad*, glutamate decarboxylase enzyme gene; NMR, nuclear magnetic resonance.
(TIFF)

**S8 Fig. Protection conferred by lactate supplementation of *E. coli* OP50.** All axonal categories of *mec-4d* animals feeding on *E. coli* OP50 supplemented with lactate. The underlying numerical data and statistical analysis for each figure panel can be found in S1 and S2 Datasets, respectively. *mec-4d*, mechanosensory abnormality protein 4.
(TIFF)

**S9 Fig. Standardization curve for GABA concentration.** Known GABA concentration plotted against absorbance values creates a curve for later estimation of GABA in samples. The underlying numerical data can be found in S1 Dataset. GABA, γ-aminobutyric acid.
(TIFF)

**S10 Fig. All morphological categories of neuroprotection in animals treated with dsRNA for GABA effectors.** Complete morphological categories of *mec-4d* animals feeding on *E. coli* HT115 expressing dsRNA for GABA effector systemically (A) and touch neuron autonomously (B). The underlying numerical data and statistical analysis for each figure panel can be found in S1 and S2 Datasets, respectively. dsRNA, double-stranded RNA; GABA, γ-aminobutyric acid; *mec-4d*, mechanosensory abnormality protein 4.
(TIFF)

**S11 Fig. Effect of DAF-2 down-regulation on neuronal degeneration in animals fed *E. coli* OP50.** (A–C) Neuronal integrity of AVM (A), ALM (B), and PLM (C) neurons of *daf-2(ts)*; *mec-4d* animals fed OP50. (D–E) All axonal categories of *daf-16; mec-4d* animals fed *E. coli* OP50 (D) and HT115 (E). The underlying numerical data and statistical analysis for each figure panel can be found in S1 and S2 Datasets, respectively. ALM, anterior lateral microtubule; AVM, anterior ventral microtubule; *daf-2*, codes for insulin-like growth factor 1 (IGF-1)

receptor; *daf-16*, ortholog of the Forkhead box transcription factor; *mec-4d*, mechanosensory abnormality protein 4; PLM, posterior lateral microtubule.
(TIFF)

**S1 Table. 1H NMR resonance assignments.** *E. coli* HT115 and HT115 Δ*gad* mutant extracts. D, doublet; dd, double doublet; *gad*, glutamate decarboxylase enzyme gene; m, multiplet; NMR, nuclear magnetic resonance; S, singlet; t, triplet.
(XLSX)

**S2 Table. 1H NMR resonance assignments.** Neuroprotective (HT115) and no neuroprotective (OP50 and HT115) bacterial extracts. D, doublet; dd, double doublet; m, multiplet; NMR, nuclear magnetic resonance; S, singlet; t, triplet.
(XLSX)

**S3 Table. Primers used for the construction of the *E. coli* HT115 Δ*gadB*::*cat*/Δ*gadA*::*kan* double mutant strain.** *gad*, glutamate decarboxylase enzyme gene. *cat*, chloramphenicol resistance gene; *kan*, kanamycin resistance gene.
(XLSX)

**S1 File. Genomics data showing unique genes of *E. coli* OP50 and HT115.**
(XLSX)

**S2 File. Differential expression analysis for all shared genes between *E. coli* OP50 and HT115.**
(XLSX)

**S3 File. MDAR.** Contains sources of experimental materials, strains, and procedures used in this research. MDAR, Materials Design Analysis Reporting.
(DOCX)

**S1 Dataset. All data with replicas from the experiments shown in the manuscript.** Numerical data for every experiment on the manuscript are contained in this dataset. Data for each figure are contained in independent tabs.
(XLSX)

**S2 Dataset. All statistical analysis of the experiments contained in the manuscript.** Statistical and post hoc analysis for every experiment on the manuscript is contained in this dataset. Data for each figure are contained in independent tabs.
(XLSX)

## Acknowledgments

We thank Irini Topalidou and Inés Carrera for critical reading of the manuscript and Diego de Mendoza for providing laboratory space and reagents. We are especially grateful to Ana Maria Pozo, who made possible the timely acquisition of key reagents. Some strains were provided by the CGC, which is funded by NIH Office of Research Infrastructure Programs (P40OD010440).

## Author Contributions

**Conceptualization:** Paula Burdisso, Andrea Calixto.

**Funding acquisition:** Andrea Calixto.

**Investigation:** Arles Urrutia, Víctor A. García-Angulo, Andrés Fuentes, Mauricio Caneo, Marcela Legüe, Sebastián Urquiza, Scarlett E. Delgado, Juan Ugalde, Paula Burdisso, Andrea Calixto.

**Methodology:** Arles Urrutia, Víctor A. García-Angulo, Paula Burdisso, Andrea Calixto.

**Writing – original draft:** Andrea Calixto.

**Writing – review & editing:** Víctor A. García-Angulo, Paula Burdisso, Andrea Calixto.

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
