## [Editor Report · Decision Letter 0]

23 Jul 2019

Dear Dr Calixto, 

Thank you for submitting your manuscript entitled "Bacterially produced GABA protects neurons from degeneration" for consideration as a Research Article by PLOS Biology.

Your manuscript has now been evaluated by the PLOS Biology editorial staff, as well as by an Academic Editor with relevant expertise, and I am writing to let you know that we would like to send your submission out for external peer review. Please accept my apologies for the long delay in sending this initial decision to you.

Before we can send your manuscript to reviewers, we need you to complete your submission by providing the metadata that is required for full assessment. To this end, please login to Editorial Manager where you will find the paper in the 'Submissions Needing Revisions' folder on your homepage. Please click 'Revise Submission' from the Action Links and complete all additional questions in the submission questionnaire.

**Important**: Please also see below for further information regarding completing the MDAR reporting checklist. The checklist can be accessed here: https://plos.io/MDARChecklist

Please re-submit your manuscript and the checklist, within two working days, i.e. by Jul 25 2019 11:59PM.

Kind regards,

Gabriel Gasque, Ph.D.,

Senior Editor

PLOS Biology

INFORMATION REGARDING THE REPORTING CHECKLIST:

PLOS Biology is pleased to support the "minimum reporting standards in the life sciences" initiative (https://osf.io/preprints/metaarxiv/9sm4x/). This effort brings together a number of leading journals and reproducibility experts to develop minimum expectations for reporting information about Materials (including data and code), Design, Analysis and Reporting (MDAR) in published papers. We believe broad alignment on these standards will be to the benefit of authors, reviewers, journals and the wider research community and will help drive better practise in publishing reproducible research. 

We are therefore participating in a community pilot involving a small number of life science journals to test the MDAR checklist. The checklist is intended to help authors, reviewers and editors adopt and implement the minimum reporting framework. 

IMPORTANT: We have chosen your manuscript to participate in this trial. The relevant documents can be located here:

MDAR reporting checklist (to be filled in by you): https://plos.io/MDARChecklist

**We strongly encourage you to complete the MDAR reporting checklist and return it to us with your full submission, as described above. We would also be very grateful if you could complete this author survey:

https://forms.gle/seEgCrDtM6GLKFGQA

Additional background information:

Interpreting the MDAR Framework: https://plos.io/MDARFramework

Please note that your completed checklist and survey will be shared with the minimum reporting standards working group. However, the working group will not be provided with access to the manuscript or any other confidential information including author identities, manuscript titles or abstracts. Feedback from this process will be used to consider next steps, which might include revisions to the content of the checklist. Data and materials from this initial trial will be publicly shared in September 2019. Data will only be provided in aggregate form and will not be parsed by individual article or by journal, so as to respect the confidentiality of responses. 

Please treat the checklist and elaboration as confidential as public release is planned for September 2019.

We would be grateful for any feedback you may have.

---

## [Decision Letter · Decision Letter 1]

23 Aug 2019

Dear Dr Calixto,

Thank you very much for submitting your manuscript "Bacterially produced GABA protects neurons from degeneration" for consideration as a Research Article at PLOS Biology. Your manuscript has been evaluated by the PLOS Biology editors, by an Academic Editor with relevant expertise, and by three independent reviewers. You will note that reviewer 1, François Leulier, signed his comments.

In light of the reviews (below), we will not be able to accept the current version of the manuscript, but we would welcome resubmission of a much-revised version that takes into account the reviewers' comments. We cannot make any decision about publication until we have seen the revised manuscript and your response to the reviewers' comments. Your revised manuscript is also likely to be sent for further evaluation by the reviewers.

Your revisions should address the specific points made by each reviewer. Having discussed these comments with the Academic Editor, we think you should add more data and analyses where the reviewers have indicated that these are necessary to clarify and solidify your conclusions. Particularly, as pointed out by all reviewers, you should especially clarify how the identified GABA dependent mechanisms relate to the IIS/FOXO signaling mechanisms identified in the manuscript. At this stage, it is not clear if GABA acts through IIS/FOXO or if the action of the HT115 strain on IIS/FOXO is a parallel and GABA independent mechanism.

In addition, while we appreciate that confirming your findings in additional models of ageing- or disease-related neurodegeneration, as suggested by reviewer 3 (major point 1), would definitively broaden the scope and appeal of your study, we do not think they are necessary for this paper. Thus, while we will welcome those data if you wish to add them, we will not press for them for further consideration in our journal. 

Please submit a file detailing your responses to the editorial requests and a point-by-point response to all of the reviewers' comments that indicates the changes you have made to the manuscript. In addition to a clean copy of the manuscript, please upload a 'track-changes' version of your manuscript that specifies the edits made. This should be uploaded as a "Related" file type. You should also cite any additional relevant literature that has been published since the original submission and mention any additional citations in your response. 

Before you revise your manuscript, please review the following PLOS policy and formatting requirements checklist PDF: http://journals.plos.org/plosbiology/s/file?id=9411/plos-biology-formatting-checklist.pdf. It is helpful if you format your revision according to our requirements - should your paper subsequently be accepted, this will save time at the acceptance stage.

Please note that as a condition of publication PLOS' data policy (http://journals.plos.org/plosbiology/s/data-availability) requires that you make available all data used to draw the conclusions arrived at in your manuscript. If you have not already done so, you must include any data used in your manuscript either in appropriate repositories, within the body of the manuscript, or as supporting information (N.B. this includes any numerical values that were used to generate graphs, histograms etc.). For an example see here: http://www.plosbiology.org/article/info%3Adoi%2F10.1371%2Fjournal.pbio.1001908#s5.

For manuscripts submitted on or after 1st July 2019, we require the original, uncropped and minimally adjusted images supporting all blot and gel results reported in an article's figures or Supporting Information files. We will require these files before a manuscript can be accepted so please prepare them now, if you have not already uploaded them. Please carefully read our guidelines for how to prepare and upload this data: https://journals.plos.org/plosbiology/s/figures#loc-blot-and-gel-reporting-requirements.

Upon resubmission, the editors will assess your revision and if the editors and Academic Editor feel that the revised manuscript remains appropriate for the journal, we will send the manuscript for re-review. We aim to consult the same Academic Editor and reviewers for revised manuscripts but may consult others if needed.

We expect to receive your revised manuscript within two months. Please email us (plosbiology@plos.org) to discuss this if you have any questions or concerns, or would like to request an extension. At this stage, your manuscript remains formally under active consideration at our journal; please notify us by email if you do not wish to submit a revision and instead wish to pursue publication elsewhere, so that we may end consideration of the manuscript at PLOS Biology.

When you are ready to submit a revised version of your manuscript, please go to https://www.editorialmanager.com/pbiology/ and log in as an Author. Click the link labelled 'Submissions Needing Revision' where you will find your submission record. 

Sincerely,

Gabriel Gasque, Ph.D., 

Senior Editor

PLOS Biology

Reviewer remarks:

Reviewer #1, François Leulier: This is a timely paper by Urrutia et al., exploring neurodegeneration in a C. elegans model, and the effect of the microbial environment in the observed phenotype. The paper is well written and contains an enormous amount of data, which seem to confirm the involvement of a microbial metabolite to protect host neurons from degeneration.

The main conclusions of the paper are:

• That E. coli strain HT115 specifically protects from a deleterious phenotype in a worm model of neuronal degeneration;

• That these effects persist with a UV-killed bacterial culture;

• That protective effects occur between 6 and 12 hours after hatching;

• That DAF-16 signaling is involved in the protective effect of E. coli HT115;

• That E. coli HT115 produces the neurotransmitter GABA, which is identified as the main molecule protecting from neurodegeneration.

Overall the paper is clear and easy to read. The controls are well chosen, and the observed phenotypes seem very clear. This article complements the current state of knowledge about bacterial metabolites impacting neuronal processes in the host and would be most welcome in the field of microbiota research.

I have however a few comments, which in my opinion can be quickly addressed by experimental data or by adding a few sentences in the discussion.

MAJOR COMMENTS:

• While in Figure 1B, P. aeruginosa PAO1, a mild pathogen, seems to confer protective effects similar to those of E. coli HT115, the only data we have on this strain is about GAD activity in Figure S5. Other soil strains are studied in Figure 7. In my opinion, if authors unequivocally want to prove the effect of GABA-producing bacteria, at least one the experiences should be performed with live or UV-killed P. aeruginosa PAO1, or if possible, with a ΔGad mutant.

• In the same manner, I think that priming with UV-killed E. coli HT115, as performed in Figure 4, should be addressed. If the metabolite is secreted and there is no nutritional effect, I assume that priming effect should be the same with live or dead bacteria.

• DAF-16 signaling is identified as a main pathway leading to neuroprotection. Unfortunately, this only comes in one figure. I think that, if documented, interactions between DAF-16 signaling and GABA should at least be addressed in the discussion. Moreover, the paper would gain strength if translocation of DAF-16 was absent in E. coli HT115 ΔGad.

MINOR COMMENTS:

• I am no expert in C. elegans. For a broad audience, such as that targeted by this journal, I think that a small introduction on mec-4 and its use as a neurodegeneration model would be extremely useful for the comprehension of the paper.

• C. elegans is always a particular model for gut microbiota research because it is a bacterivore organism, which feeds on bacteria present on the environment. Authors claim that the effects of E. coli HT115 are not nutritional, based on the effect of UV-killed bacteria. In a physiological soil environment, how to separate nutritional from non-nutritional effects? I think this could be addressed in the discussion.

• Are the strains used in Figure 1 commonly used for C. elegans research? The pertinence of the strains should be explained for non-specialists of C. elegans.

• Authors focus on axon growth and degeneration. I wonder if these effects are simply the result of growth and degeneration of neuronal processes, or if it could involve processes of neurogenesis and neuronal apoptosis. How to distinguish between these situations?

Reviewer #2: The work by Urritia and colleagues explores the role of bacteria in a C. elegans neurodegeneration model. The authors perform a vast array of functional and physiological assays in both C. elegans and bacteria to uncover the role of the bacterial metabolite GABA as a key molecule in regulating neuronal decay. While that finding is not entirely unexpected, this work does a great job at highlighting, beyond reasonable doubt, the key role of GABA in regulating this important phenotype. Importantly, the findings of the distal action of GABA on neuronal function are novel, exciting and have wide-relevance for potential biomedical applications. 

The work presented here supports very well all the conclusions put forward by the authors. Importantly, the experimental approaches used are sound and well described, allowing for full replicability of the results.

Points to be addressed:

1) The title is too broad since it is not known whether this is an evolutionarly conserved phenomena. It should be clearly stated that these findings were obtained in C. elegans.

2) The authors should show that GABA levels and GAD enzymatic levels from all bacterial strains tested here and with HT115 at different growth times correlate with the worm phenotypes and perform appropriate statistical analysis to show the power of that correlation.

3) The authors should show which tissue(s) are important for the role of daf-16 in regulating the neuronal phenotypes.

4) The authors should attempt to discuss the connection between GABA and the IIS.

5) It is intriguing why HT115 is rewired to produce GABA more than other E. coli strains. Could the authors comment on this based on their genomic and transcriptomic data? It would be interesting to see what happens to this worm phenotype when cultured in an OP50 strain where ssa is deleted.

6) metabolites are not over-expressed. Change to over-abundant.

7) Please correct E. coli K12 or K 12 to K-12. Also, please clarify that HT115 is a K-12 strain and OP50 is a B strain. Finally, state what is the K-12 and B strain used in this study.

Reviewer #3: In this manuscript, the authors use the degeneration of C. elegans touch receptor neurons in mec-4(gf) animals as a model to investigate the potential effects of dietary bacteria on necrosis/neurodegeneration during development. They report that some laboratory and environmental bacterial strains exhibit more protective effects against degeneration over others. In particular, among the strains tested, E. Coli HT115 can significantly prevent or protect touch receptor neurons and others in mec-4(gf) and deg-1 animals when compared to E. Coli OP50, which is routinely used for maintaining and culturing C. elegans under laboratory conditions. They further uncover that bacterially produced GABA appears to be critical for the neuronal protection function of HT115. 

The relationship between gut microbiota and brain function has been a hot topic , and some recent studies highlighted the potential link between gut microbiota and certain neurodegenerative diseases. While the paper does nice job in illustrating the protection effects of HT115 in the mec-4(gf) degeneration model, the paper falls short in dissecting the mechanism of this protection as well as addressing the broad interest of this discovery. 

Major points

1) As mentioned in this manuscript, mec-4(gf) caused necrosis of TRNs through Na+ entry, Ca++ influs, and ROS imbalance. Does the protection effect of HT115 acts through suppressing any of them? if so, is GABA a mediator for this regulation. The mec-4(gf) and deg-1 both cause neuronal degeneration through necrosis, and the degeneration is relevant to their channel functions, which is a very special type of neuodegeneration and may not be relevant to the common causes of neurodegeneration, such as ageing, neuronal injury, and neurodegenerative diseases. Therefore, the conclusions of this study may not been applicable to a broad concept of neurodegeneration. TNR neurons display degeneration during ageing, and there are some established neurodegenerative disease models in C. elegans. It is worth to confirm the board impact of bacterially produced GABA in ageing or disease related neurodegeneration. 

2) The study identifies the difference between HT115 and OP50 that renders the neuroprotective effects, which turns out to be GABA, but does not fully explore the mechanisms of dietary uptake of GABA in promoting neuronal health. The authors did show that GABA receptors and related transporters were involved somewhere outside of touch receptor neurons, but where possibly could it be? It is necessary to know which tissues/cells are the primary targets of GABA, and this data is also essential in understanding how GABA functions in HT115 related neuronal protection. In this manuscript, all analyses of GABA receptors/transports were down using RNAi, and it should be conformed by loss-of-function mutants of those genes as well as tissue special rescue experiments. 

Most of GABA functions are tightly coupled with its concentrations, but it doesn’t seem to be the case in this study. As shown in Fig. 7, in O.P. 50 the GAD enzyme activity and the ability of producing GABA are nearly 50% of that in HT115, but O.P. 50 has no protect effect. In contrast, in Fig.2, 1/100 (1%) dilution of HT115 in O.P. 50, in which the GABA concentration is likely 50%+1% of HT115’s, have almost 50% of protection effects of 100% HTT115. It is difficult to understand that how 1% additional HT115 GABA in O.P. 50, which is already 50% of HT115 GABA, causes such significant changes of phenotypes. To me, it suggests that something else in HT115 may be more important than GABA for its protection effects. 

The data in Fig. 4 is also difficult to understand, that feeding the animals with HT115 before AVM are born can protect them from degeneration. Does this suggest mec-4(gf) is already expressed and cause the degeneration of the Q lineage before AVM are born? Or HT115 causes a long lasting change, which protects AVM after they are born. If it is the later case, GABA may not be the reason to cause HT115 protection, as GABA neurons are continually generating GABA but failed to protect AVM with O.P.50 feeding. 

In Fig. 5 it seems DAF-16::GFP only has different in 24 h after hatch when feeding with o.p.50 and h115. Does this mean daf-16 is not required for the protecting effects mentioned in Fig.4 when feeding worms only in the first 12 h after hatch

3) The authors also brought DAF-2-DAF-16 signaling into the equation, but it is not genetically clear in what fashion this insulin-like signaling is involved in dietary GABA’s neuroprotective function. daf-2(lf) cause constitutive activity of daf-16, but daf-2(lf) didn’t protect AVM in O.P. 50 feeding (Fig. 5), which suggests that daf-16 activation is not sufficient to protect AVM from degeneration. If that is case, the DAF-16::GFP expression change in Fig. 5D may not be relevant to the phenotype. For the data shown in Fig. 5E-F, it shows daf-16 is involved in AVM degeneration but not necessary involved in HT115 protection. 

4) HT115 feeding may suppress mec-4(gf) expression to protect the cause of AVM degeneration. It should be straightforward to test this by RT-PCR or in vivo labeling of mec-4. 

Minor points

1) It will be good to test whether feeding mec-4(gf) with different bacteria affects the its lifespan.

2) Fig. 5D, was the nuclear DAF-16::GFP expression assessed based on whole animals or was it neuron-specific? It would be helpful to indicate it in the Methods or figure legends.

3) Do endogenous GABA expression levels differ between widl-type animals and mec-4d mutants during larval stages? There is a very good GABA antibody for worms, and it should easy to be tested 

4) OP50 + pGadA + glutamic aid can protect neurons from degeneration, but not quite comparable to HT115 or HT115 + Gad mutation + pGadA

5) Fig. 1E, statistical significance?

---

## [Decision Letter · Decision Letter 2]

9 Jan 2020

Dear Dr Calixto,

Thank you for submitting your revised Research Article entitled "Bacterially produced metabolites protect C. elegans neurons from degeneration" for publication in PLOS Biology. I have now obtained advice from the original reviewers and have discussed their comments with the Academic Editor. You will note that reviewer 2, Filipe Cabreiro, has signed his comments. 

Based on the reviews, we will probably accept this manuscript for publication, assuming that you will modify it to address the remaining points raised by reviewer 2 and by us and the Academic Editor. 

Having discussed reviewer 2’s point 1 with the Academic Editor, we do not think you should necessarily address it experimentally. However, while we agree that many of the originally posted questions and doubts which have not been addressed go beyond the current scope of the manuscript, it would be good if the you could add some few sentences to the Discussion to critically discuss other interpretations of how GABA and lactate production could mediate neuroprotection. Like the reviewers, the Academic Editor thinks you have shown that GABA production is required for neuroprotection, but the mechanism is less clear. For example, it seems that GABA only acts in the context of bacteria being present; therefore, the story is more complicated than presented. The Academic Editor also thinks that, for example, one likely possibility is that GABA and lactate to a large extent actually act within/on the bacteria to allow them to do whatever they do to the host. Another unsolved question is the lack of effect on DAF-16 localization with GABA supplementation. Therefore, we think you should be more critical of your findings and discuss the limitation of your data and alternative interpretations to be addressed in future research. 

We expect to receive your revised manuscript within two weeks. Your revisions should address the specific points made by each reviewer and by the editors. Please submit the following files along with your revised manuscript:

***IMPORTANT: Please also make sure to address the data and other policy-related requests noted at the end of this email.

In addition to the remaining revisions and before we will be able to formally accept your manuscript and consider it "in press", we also need to ensure that your article conforms to our guidelines. A member of our team will be in touch shortly with a set of requests. As we can't proceed until these requirements are met, your swift response will help prevent delays to publication.

*Copyediting*

*Published Peer Review History*

*Early Version*

*Submitting Your Revision*

Sincerely,

Gabriel Gasque, Ph.D., 

Senior Editor

PLOS Biology

DATA POLICY:

Please ensure that the figure legends in your manuscript include information on where the underlying data can be found, and ensure your supplemental data file/s has a legend.

Reviewer remarks:

Reviewer #1, François Leulier: The authors replying in a satisfactory manner to my previous comments and I feel the manuscript is now ready for publication.

Reviewer #2, Filipe Cabreiro: The authors addressed all my previous points.

I have a remaining point as a result of their new data and discussion which I think needs to be addressed textually or experimentally if they would like to further support their claims.

Minor points:

1) I don't fully agree with the discussion provided to support their point related to the low production of GABA in B strains. While the rcs may be lost in the B lineage, GABA is still being detected at low levels in OP50. The authors data shows this - approximatelly 50uM, which is far from being neglectable. Therefore, I don't think the authors can say that OP50 lost its capacity to produce GABA. Also, the authors don't explain why HT115 is a particularly high producer of GABA (approximatelly 8-fold higher) when compared to BW25113 which is another derivative of the parental MG1665 K-12 strain. Is this an unexpected trait that only happened to a very highly modified laboratory strain? How does this compare to the parental "wild-type" K-12 strains MG1655 and W3110. The authors should address this experimentally if possible, or alter their discussion to convey their findings more accurately.

2) There is a misspelling in rcsB (the authors named it rscB and rsc) and the nomenclature for K-12 isn't always consistent (sometimes shown as K12).

Filipe Cabreiro

Reviewer #3: The revised version of this manuscript is great improved. I especially appreciate the new experiments adding Lactate in this context. I recommend the manuscript for publication

---

## [Editor Report · Decision Letter 3]

18 Feb 2020

Dear Dr Calixto,

On behalf of my colleagues and the Academic Editor, Carlos Ribeiro, I am pleased to inform you that we will be delighted to publish your Research Article in PLOS Biology. 

PRESS 

Kind regards,

Alice Musson

Publication Assistant, 

PLOS Biology

on behalf of

Gabriel Gasque,

Senior Editor

PLOS Biology